# Genomic Characterization and Initial Insight into Mastitis-Associated SNP Profiles of Local Latvian *Bos taurus* Breeds

**DOI:** 10.3390/ani13172776

**Published:** 2023-08-31

**Authors:** Dita Gudra, Anda Valdovska, Daina Jonkus, Daiga Galina, Daina Kairisa, Maija Ustinova, Kristine Viksne, Davids Fridmanis, Ineta Kalnina

**Affiliations:** 1Latvian Biomedical Research and Study Centre, LV-1067 Riga, Latvia; dita.gudra@biomed.lu.lv (D.G.); maija.ustinova@biomed.lu.lv (M.U.); kristine.viksne@biomed.lu.lv (K.V.); davids.fridmanis@biomed.lu.lv (D.F.); 2Faculty of Veterinary Medicine, Latvia University of Life Sciences and Technologies, LV-3001 Jelgava, Latvia; 3Scientific Laboratory of Biotechnology, Latvia University of Life Sciences and Technologies, LV-3001 Jelgava, Latvia; 4Faculty of Agriculture, Latvia University of Life Sciences and Technologies, LV-3001 Jelgava, Latviadaina.kairisa@lbtu.lv (D.K.); 5Scientific Laboratory of Molecular Genetics, Riga Stradins University, LV-1007 Riga, Latvia

**Keywords:** Latvian Blue cow breed, Latvian Brown cow breed, local endangered cow breeds, whole genome sequencing, mastitis

## Abstract

**Simple Summary:**

Cows are one of the oldest livestock that are bred for both meat and milk. There are only two local cow breeds in Latvia—Latvian Brown and Latvian Blue—which, until now, have not been genetically characterized. This study examined the genetic background of both local breeds in Latvia and identified genetic factors associated with mastitis, a common inflammation of the mammary gland that affects animal welfare and the quality of milk. Whole genome sequencing was performed on blood and semen samples, and the results from the obtained data show that both breeds had similar levels of genetic diversity compared with other breeds. The Latvian Blue breed showed a higher genetic variance compared with the Latvian Brown breed. Specific mastitis-associated genome regions were identified for each breed. This study provides fundamental insights into the genetics of local Latvian cow breeds and provides the first insight into the hereditary factors of mastitis in dairy cows. It may also facilitate the development of genetic tests to aid breeders in their attempts to maintain and improve local and foreign Latvian cattle breeds.

**Abstract:**

Latvia has two local *Bos taurus* breeds—Latvian Brown (LBG) and Latvian Blue (LZG)—characterized by a good adaptation to the local climate, longevity, and high fat and protein contents in milk. Since these are desired traits in the dairy industry, this study investigated the genetic background of the LBG and LZG breeds and identified the genetic factors associated with mastitis. Blood and semen samples were acquired, and whole genome sequencing was then performed to acquire a genomic sequence with at least 35× or 10× coverage. The heterozygosity, nucleotide diversity, and LD analysis indicated that LBG and LZG cows have similar levels of genetic diversity compared to those of other breeds. An analysis of the population structure revealed that each breed clustered together, but the overall differentiation between the breeds was small. The highest genetic variance was observed in the LZG breed compared with the LBG breed. Our results show that SNP rs721295390 is associated with mastitis in the LBG breed, and SNPs rs383806754, chr29:43998719CG>C, and rs462030680 are associated with mastitis in the LZG breed. This study shows that local Latvian LBG and LZG breeds have a pronounced genetic differentiation, with each one suggesting its own mastitis-associated SNP profile.

## 1. Introduction

The ‘wealth’ of a country can be measured by its animal breeds that are bred to facilitate the survival of resident populations within specific environmental conditions. The Convention on Biological Diversity recommends the conservation of different local breeds, as well as the related agricultural systems, as an optimal strategy for preserving genetic variation for further use in livestock development, thus emphasizing the need to promote ex situ conservation and the sustainable use of local genetic resources [1,2].

Latvia is home to two indigenous dairy cow breeds, Latvian Brown old type or ‘Latvijas brūnā govs’ old type (LBG), which, according to 2023 data, consists of 180 cows and 62 heifers, and Latvian Blue or ‘Latvijas zilā govs’ (LZG) with a population of 390 animals. Since 2000, the Ministry of Agriculture of Latvia has been actively involved in conservation efforts, implementing approved breeding programs for endangered varieties, like LBG and LZG cows. As of 2018, the conservation of local endangered varieties in accordance with the ‘endangered breed’ definition has been guided by Article 2 Point 24 of the Regulation (EU) 2016/1012. This regulation addresses zootechnical and genealogical conditions for the breeding, trade, and entry of purebred breeding animals, hybrid breeding pigs, and germline products into the European Union. It also includes amendments to Regulation (EU) No. 652/2014 and Council Directives 89/608/EEC and 90/425/EEC, while repealing certain acts in the area of animal breeding [3]. According to this definition, an ‘endangered breed’ refers to a local breed that is recognized as endangered by a Member State, and is genetically adapted to traditional production systems or environments within the Member State. The endangered status is scientifically established by a body possessing the necessary skills and knowledge of endangered breeds. The preparation of programs for the conservation and breeding of endangered breeds is carried out by animal breed societies [4]. The Animal Breeders Associations of Latvia developed specific programs for the conservation of endangered old-type LBG [5] and LZG [6] cows. Consequently, both cattle breeds, LBG and LZG, are recognized and listed in the European Regional Focal Point for Animal Genetic Resources (ERFP) database [7].

Domestic cows that were reared on the territory of Latvia until the second half of the 19th century had small bodies, but they were very modest and durable. These cows had a relatively lower milk yield, which, starting in 1862, resulted in the import of the Angler variety to improve the local breeds [8]. Later, at the beginning of the 20th century, Danish Red cattle became known in Europe and were also used to improve the yield and composition of milk produced by local Latvian cows. The result of this century-long breeding was solidified in 1922 when these improved local brown-haired animals were recognized as a separate breed named Latvian Brown cows with the breed mark LB. In 2004, a breed preservation program aimed at the preservation of the local LBG old-type cows was developed and included all animals with both parents descending from the LBG breed in the last four generations, with at least 50% of LBG blood, and the rest coming from Danish Red or Angler breeds [8,9,10]. The other breed—Latvian Blue (breed mark LZ)—was registered in the 1930s. Because of the small number of cows, animals of this breed were gradually assimilated with those of other breeds in Latvia. From the middle of the 20th century, the number of pure-breed animals started to dwindle; however, calves with blue–gray hair were periodically born. The restoration of the LZG was difficult because of the lack of unaffiliated breeding animals. To avoid a rapid increase in inbreeding, the Latvian association ‘Blue cow’ began using breeding animals from the Tyrol Grey and Lithuanian Light Grey breeds. Cows with the LB and LZ breed marks are well adapted to local conditions, can graze on relatively poor pastures, and are often kept in old-type animal sheds [9,10].

The erosion of genetic resources experienced by both local cattle breeds often leads to the loss of beneficial traits, such as a naturally developed resistance to common diseases [11,12]. Mastitis, or inflammation of the mammary glands, is the most common disease in the ruminant industry with significant economic, hygienic, and welfare implications, and persists in all animal production systems despite improved management practices [13,14]. Traditionally, bovine mastitis is classified as subclinical, clinical, or chronic, and designation is based on the progression of inflammation. The diagnosis of mastitis is often based on the somatic cell count (SCC), i.e., the number of inflammation cells per mL of milk; the presence of microbial pathogens; and clinical examination in more severe cases. More recently, attention has been paid to the application of metabolomics to improve diagnostic effectiveness [14,15]. Since bacterial infection is the major cause of mastitis, the pathogen type and characteristics such as virulence, biofilm formation, and antimicrobial resistance have been considered dominant determinants of the disease course [13,16]. The development of mastitis is also affected by interactions among factors such as the cattle housing conditions, milking routine, cow breed, body condition, udder morphology, cow age, lactation stage, and productivity [11,12,13,14,17,18]. Differences in mastitis resistance among cattle breeds support the hypothesis that genetic components modulate the risk for the disease [11,12,18]. Currently, more than two thousand potential candidate loci have been linked to mastitis. The most potent candidate genes for mastitis susceptibility and resistance are involved in regulating immune-system-related processes. Others are associated with milk-production-related traits, including the milk yield and composition of the milk, as well as reproduction-related traits [19,20,21]. Some candidate genes have been linked with body conformation traits, like the hip conformation of legs, and muscle-related traits along with meat quality [19]. However, the heritability estimates for mastitis and mastitis-related traits are generally low. For example, the heritability of clinical mastitis ranged from 0.04 in Red Dairy cattle to 0.05 in Holstein, whereas the heritability estimates for SCC in these breeds were 0.12 and 0.13, respectively [22]. However, it was also observed that there was a higher variation in the heritability for SCC (between 0.07 and 0.24) depending on the lactation period [23]. Therefore, it cannot be excluded that the heritability of mastitis-related traits has been underestimated because of the complex nature of genetic backgrounds [20,24].

The application of traditional selection approaches, which are based on phenotype and pedigree data, help to reduce the incidence of mastitis. Recently, it has become possible to complement traditional approaches (based on yield, health, and exterior indices) with breeding values estimated during genomic analyses. Since the cost of sequencing continues to decline, the number of studies based on whole genome sequencing is gradually increasing, and the accumulation of information on genetic variation that underlines disease resistance and other desirable traits can improve breeding programs and prevent the co-segregation of undesirable traits [25]. Therefore, this study aims to employ a whole-genome sequencing approach to genetically characterize two local Latvian breeds, LBG and LZG, and identify the genetic factors that determine resistance against mastitis. To the best of our knowledge, this is the first in-depth genetic characterization of local Latvian dairy cattle breeds.

## 2. Materials and Methods

### 2.1. Selection of Animals and Sample Collection

In Latvia, cows of local origin are mainly reared in small individual farms; the exception is the teaching and research farm Vecauce, with around 50 LBG old-type cows. In the summer season (from May to October), cows are grazed, while in the winter, the dairy cattle are housed in tied stalls and fed with silage and hay. The cows are milked twice a day using portable or semi-automatic milking machines. Similar hygiene management practices were maintained across all herds. The health management and control of the herds included in the study were conducted as part of routine veterinary surveillance. Impregnation of animals is primarily achieved through artificial insemination using semen from artificial insemination stations (AIS). Therefore, all employed blood samples from cows (5 mL of blood within collection BD Vacutainer^®^ EDTA Tubes (Beckton-Dickenson, Franklin Lakes, NJ, USA)) were acquired by a qualified veterinarian during routine general health checkups on the animals, which involved blood sampling. Kurzemes and Siguldas AIS kindly provided the employed bull semen samples.

The selection of animals for blood and semen sampling was based on pedigree information available from the Agricultural Data Center (ADC). The sample set included a total of 80 animals comprising 36 LZG cows, 36 LBG cows, and 8 bulls (four from each breed). Deep sequencing analysis was performed for 16 animals representing dominant paternal lineages in LBG and LZG breeds. Four unrelated bulls were selected from each breed, representing dominant paternal lineages with >60% ancestry from the respective breed. All bulls were born after 2010. The final decision to include a particular animal was based on the availability of bioproducts from the AIS resources.

For the LBG breed, four unrelated cows (Amula, Bruka, Duda, and Guste) and four unrelated bulls (Brinums, Namejs, Vilsons, and Railis Ullors) were from Odins, Rudme, Potrimps, and Ullors paternal lineages. These were previously used in the conservation of the breed. The 32 cows included in the shallow genome sequencing analysis were related according to the respective dominant lineages—the sample set included 6 daughters of bulls of the Rudme lineage, 6 of the Odins lineage, 7 of the Potrimps lineage, and 13 of the Ullors lineage (Appendix A). LBG animals included in the study came from six different herds (Appendix A) with similar hygiene management practices.

The LZG breed has, in general, a higher kinship connection; therefore, four LZG bulls (Aizups Lietuvietis, Karlos, Pipars, and Samts) and four cows (Darlinga, Dzesika, Kripatina, and Saulite) were selected from different herds for deep genome sequencing. These animals represented only two paternal lineages, Gaujars Lietuvietis and LBG Potrimps. Thirty-two cows of the LZG breed from four farms were selected for shallow genome sequencing and were related according to their dominant lineages; samples were taken from ten cows of the Potrimps bull lineage, seven cows of the Gaujars bull lineage, three daughters of the Dzilnis lineage, and seven cows of the Darbonis lineage, and four samples were taken from two bulls with Lithuanian Light Grey ancestry related to Darbonis, and one sample was taken from a TYG bull (Appendix A). A graphical representation of kinship among animals included in Appendix A, along with dominant paternal lineages for both breeds, was created with Gephi v.0.10.1. [26]. Cows of the LZG breed were selected from eight herds (Appendix A) with similar hygiene management practices.

### 2.2. Milk Sample Collection and Bacteriological Testing

Milk samples were obtained from 28 LBG cows and 21 LZG cows kept in six and seven herds, respectively (Appendix A). These were then analyzed on a quarterly basis (elevated SCC in milk count controls above 200,000 SCC/mL) and compiled with previous data. An increased SCC (>200,000 SCC/mL) was classified as an indicator of subclinical mastitis. Samples were collected aseptically (as proposed by [27]) in sterile tubes (~10 mL per quarter) prior to milking and stored at 4 °C until delivered to the laboratory within 4 h.

Subclinical mastitis is defined as the presence of inflammation within the normal-appearing mammary gland, but the diagnosis of an intramammary infection is based on the identification of the infectious agent. To isolate bacteria, milk samples (100 µL) were spread aseptically on Columbia blood agar base with 5% defibrinated blood and MacConkey Agar (Oxoid, Basingstoke, UK). Incubation was carried out at 36 ± 1 °C for 24–48 h. Bacterial isolates were identified via culture growth (recommended by [27]), morphology (Gram stain), and biochemistry with Vitek MS Matrix-Assisted Laser Desorption Ionization Time-of-Flight (MALDI-TOF) technology, according to the manufacturer’s (bioMerieux SA, Craponne, France) recommendations.

### 2.3. DNA Extraction

Prior to extraction, acquired blood samples were centrifuged for 15 min at 3488× *g* and 4 °C. Genomic DNA was then extracted from the buffy coat using the phenol–chloroform method following a previously established routine DNA extraction protocol [28].

A custom genomic DNA extraction protocol was designed for semen samples. Semen samples were mixed with 10 mL of buffer (150 mM NaCl and 10 mM EDTA (pH 8.0)) in a 50 mL tube and vortexed for 10 s. Next, buffered samples were centrifuged for 10 min at 3488× *g* at room temperature, and the supernatant was removed, leaving the pellet and approx. 1 mL of buffer. The pellet was then suspended by vortexing for 10 s, and the resulting suspension was transferred to a clean 2.0 mL microcentrifuge tube. An additional 0.5 mL of buffer was added to the same 50 mL tube, vortexed for 10 s to collect any semen that adhered to the walls of the tube, and the suspension was transferred to the same microcentrifuge tube. Samples were then centrifuged for 2 min at 14,100× *g*, and the supernatant was discarded. Next, pellets were resuspended in a 300 μL buffer containing 100 mM TrisHCl (pH 8.0), 10 mM EDTA (pH 8.0), 500 mM NaCl, 1% SDS, and 2% β-mercaptoethanol. To release the DNA from its protamine complex, 40 μL of Proteinase K (Macherey-Nagel, Duren, Germany) was added to the buffered sample, which was mixed and incubated for 2 h in VorTemp 56 incubator (Labnet, Medley, FL, USA) with a rocking platform at 55 °C and 200 rpm. An additional 20 μL of Proteinase K was added, and samples were incubated for another 2 h on the same rocking platform at 55 °C and 200 rpm. Finally, the robotic system KingFisher Duo (Thermo Fisher Scientific, Waltham, MA, USA) combined with the NucleoMag Tissue reagent kit (Macherey-Nagel, Germany) was used to purify released DNA. The purification procedure was carried out according to the user manual section ‘Genomic DNA From Tissue’ starting from the third step. During the final purification step, DNA was eluted in 200 μL of Buffer MB6 (Macherey-Nagel, Germany).

The quality of extracted DNA was evaluated using 1.2% agarose gel electrophoresis, while the quantity was assessed using the Qubit High Sensitivity DNA assay kit and the Qubit 2.0 fluorometer (Thermo Fisher Scientific, USA).

### 2.4. Library Construction and Sequencing Analysis

Prior to the genome sequencing analysis, concentrations of DNA samples were normalized to an initial library input of 500 ng and sheared using Covaris S220 Focused-ultrasonicator (Covaris, Woburn, MA, USA), employing the manufacturer’s provided parameters to reach an average fragment size of 400 bp. Libraries were further prepared using MGIEasy Universal DNA Library Prep Set V1.0 (MGI Tech Co., Shenzhen, China) according to the manufacturer’s recommendations. The libraries’ quality control was assessed using the Qubit High Sensitivity dsDNA assay kit and the Qubit 2.0 instrument, as well as the Agilent High Sensitivity DNA kit and the Agilent 2100 Bioanalyzer (Agilent Technologies, Santa Clara, CA, USA).

Samples were then diluted and pooled to achieve at least 440 million reads per sample for deep sequencing (estimated coverage of approx. 45×) and 100 million reads per sample for shallow sequencing (estimated coverage of approx. 10×). Pooled and circularized libraries were used as templates for DNA nanoball (DNB) preparation and subsequently loaded onto the PE150 flow cell. Sequencing was carried out using the DNBSEQ-G400 sequencer and a DNBSEQ-G400RS High-Throughput Sequencing Set (MGI Tech Co., China) according to the manufacturer’s instructions.

### 2.5. Sequencing Data Processing and Variant Calling

Quality control and quality trimming of the obtained paired-end reads were performed using Trimmomatic v0.39 [29] with options LEADING:30, TRAILING:30, and a minimum read length of 36 nt. Quality-filtered reads were aligned to the *Bos taurus* reference genome ARS-UCD1.2_Btau5.0.1Y [30] using the BWA v.0.7.17 MEM algorithm [31]. Mapping results were then converted into the BAM format and sorted based on coordinates using GATK [32] SortSam. PCR duplicate reads were marked using GATK MarkDuplicates, and base quality scores were recalibrated using GATK BaseRecalibrator and ARS1.2PlusY_BQSR_v3.vcf.gz reference file of known polymorphic sites. Then, base quality score recalibration was applied to the sample data, and haplotypes were called using GATK HaplotypeCaller [33]. All samples were combined into a single genomics database using GATK GenomicsDBImport, and joint genotyping was performed using GATK GenotypeGVCFs. Identified variants were annotated using the SnpEff tool [34] and the ARS-UCD1.2.99 database. Hard filtering was applied using GATK VariantFiltration to exclude potential false positive variant calls with the following parameters: QD < 2.0; ReadPosRankSum < −8.0; FS > 60.0; MQ < 50.0; SOR > 3.0; MQRankSum < −12.5; QUAL < 30; and DP > 20. Variants that passed hard filtering were selected for downstream analysis using Bcftools [35], and each passed variant was assessed for Hardy–Weinberg equilibrium using PLINK v.1.90b6.21 [36]. Bcftools was used to calculate basic statistics of filtered joint genotype files and to estimate common and unique variants in LZG and LBG samples. For comparison with other breeds, whole genome raw sequencing data from NCBI Sequence Read Archive (SRA) were obtained for the breeds Holstein (HLS) SRR934405, SRR934414, and SRR934413; Hereford (HRF) SRR15840409, SRR15840411, and SRR15840412; Limousin (LMS) SRR8727721, SRR8727724, and SRR8727725; Belgian Blue (BBL) SRR7363725, SRR7363701, and SRR7363723; Tyrolean Gray (TYG) ERR4291969, ERR4291967, and ERR430968; and German Angus (GER) SRR6895116, SRR20012772, and SRR20012773, and were analyzed with the same parameters and genome references as the LZG and LBG breeds (Appendix A).

### 2.6. Genomic Analysis

The SNP density plot per chromosome of LBG and LZG breed was visualized using the package rMVP [37] v.1.0.6. The degree of polymorphisms on each chromosome, e.g., nucleotide divergence (π), and Tajima’s D statistic were estimated using VCFtools v.0.1.17 [38] with a 50 kb sliding window and visualized on each chromosome using the ggplot2 package in the R environment. The distribution of nucleotide divergence, Tajima’s D statistic, F_ROH_, and F_ROH>50kb_ values were assessed using the Shapiro–Wilk normality test and further used to compare their significance between LBG and LZB breeds using the Wilcoxon rank sum test at a *p*-value threshold of 0.05. Genetic differentiation (F_ST_) was estimated using VCFtools with parameters of a 40 kb sliding window and a 20 kb step size. Individual heterozygosity was calculated using VCFtools. The proportion of heterozygosity was calculated as the ratio of the number of heterozygous sites divided by the total number of sites. To compare LBG and LZG breeds to other breeds that were obtained from the NCBI SRA, the set of LBG and LZG samples was randomly reduced to three samples per breed using bcftools. An additional check was performed on randomly selected samples to ensure that they were not located close to each other in the genetic distance tree and were not related (Appendix A). Sample reductions in the LBG and LZG sample sets were performed to ensure comparability with breed samples obtained from SRA (*n* = 3 per breed). Pairwise comparisons of (i) nucleotide divergence and (ii) heterozygosity among all breeds were conducted using the T-test with the Bonferroni *p*-value adjustment method with a *p*-value threshold of 0.05 employing package rstatix v.0.7.2.999. Visualization of the results was performed using package ggpubr v.0.5.0.999. For the runs of homozygosity (ROH) analysis, only variants with a minor allele frequency above 0.05 were retained, and ROH was then calculated using PLINK with the following parameters: minimum SNP count of 100, minimum ROH length of 10 kb, and only one heterozygous call within the ROH. The average ROH in kilobases per individual per chromosome was calculated, and the results of the ROH analysis were visualized using pheatmap v.1.0.12. The package PopLDdecay v.3.42 [39] was used to estimate the linkage disequilibrium degree for all breeds, and the obtained results were visualized using the matplotlib [40] library in the Python environment. Inbreeding levels were described using measures based on runs of homozygosity (F_ROH_), calculated by dividing the sum of lengths of all ROH (F_ROH_) or the sum of lengths of ROH longer than 50 kb (F_ROH>50kb_) with the total length of autosomes (L_AUTO_; 2,489,385,779 bp) using the following [41]:F_ROH_ = L_ROH_/L_AUTO_(1)
or
F_ROH>50kb_ = L_ROH>50kb_/L_AUTO_(2)

### 2.7. Population Structure

Principal component analysis (PCA) and structure analysis were conducted to investigate the genetic background. SNPs with high linkage disequilibrium were removed using PLINK. The pruned SNP data were used to estimate individual ancestry using the maximum likelihood method implemented in ADMIXTURE v.1.3 [42]. The default parameters (folds = 5) for cross-validation and the lowest cross-validation error were considered the most likely K value. The results were visualized using the base graphics package within the R environment. To represent the genetic relatedness of animals included in the study, a genetic distance tree based on the UPGMA algorithm with 100 bootstrap replicates was built using package poppr v.2.9.3. [43].

### 2.8. Detection of Variants Associated with Mastitis

An association analysis was performed to determine whether the genetic background of the cattle was associated with mastitis resistance. First, variants were filtered using PLINK to retain only those with a minor allele frequency above 0.05. Pedigree information and disease status were collected for each animal and submitted to PLINK for association analysis. Genomic inflation estimator lambda (λ) based on the median value of the Chi-squared test was acquired through PLINK multiple-testing-corrected *p*-values. The genome-wide association results were visualized using Manhattan and quantile–quantile (Q-Q) plots with the qqman [44] package. The resulting mastitis-associated SNPs were considered significant if their *p*-value was less than 1.0 × 10^−9^ and if their Bonferroni-corrected *p*-value was less than 0.05. All the above-described computations were performed using Riga Technical University’s High Performance Computing Center infrastructure.

### 2.9. Statistical Analysis of SCC Data

Descriptive statistics, Pearson correlation, and linear regression analysis methods were performed in the Stata v.17.0 (StataCorp LP, College Station, TX, USA) software. All SCC data were converted into SCC linear score (LS) according to the following formula (Equation (3)) and expressed in log_2_ units:LS_SCC_ = ln (SCC × 10^−5^) × (ln 2)^−1^ + 3(3)

## 3. Results

### 3.1. Variant Calling and Distribution of Polymorphisms

Our strategy for assessing genetic diversity in two local cattle breeds and identifying genomic regions that determine breed-specific traits was based on a combination of deep (eight animals: four cows and four bulls) and shallow (32 cows) sequencing for animals of each breed. For LZG breed deep sequencing, we acquired a total of 3.5 billion sequencing reads with an average of 430,650,120 (min 394,550,618; max 476,190,706) and 150 bp reads per sample, roughly corresponding to a 45.02× coverage (min 41.24×; max 49.78×). For the LBG breed, a total of 3.6 billion reads were obtained, with an average of 450,985,871 (min 420,703,847; max 485,317,767) and 150 bp reads per sample, corresponding to a 47.14× coverage (min 43.98×; max 50.73×). The shallow sequencing of the LZG samples yielded a total of 3.6 billion sequences with an average of 112,412,214 reads per sample (min 88,104,565; max 172,357,510) and with an estimated average genome coverage of 11.75× (min 9.21×; max 18.02×) (Figure 1A). The shallow sequencing of the LBG samples yielded a total of 3.5 billion sequencing reads with an average of 109,351,724 reads per sample (min 51,046,361; max 158,358,455). Here, the estimated average genome coverage was 11.43× (min 5.34; max 16.55×) (Appendix A).

Variant calling on assembled sequencing data revealed that after quality filtering, there were 15,496,197 SNPs and 2,449,533 indels within the LBG data set, but out of these, 713,680 were multiallelic sites, and 50,123 were multiallelic SNP sites. Within the LZG data set, 16,248,091 SNPs and 2,519,762 indels passed the quality filtering criteria (Figure 1B), while the total number of multiallelic sites here was 740,920, and the number of multiallelic SNP sites was 54,372. A comparison of the variant repertoire within both data sets uncovered that the LZG and LBG sample sets shared 14,983,960 variants, while 1,774,684 were unique to the LBG sample set and 3,694,671 were unique to the LZG sample set. Furthermore, most of these SNPs were located within introns (LZG—14,405,912 SNPs accounting for 46.66%; LBG—13,721,383 SNPs accounting for 46.54%) and the intergenic regions (LZG—13,148,356 SNPs accounting for 42.59%; LBG—2,583,520 SNPs accounting for 42.68%) (Figure 1C). Thus, only a small proportion of SNPs were located within exons—0.83% within the genomes of the LBG breed animals (243,178 SNPs) and 0.83% within the genomes of the LZG breed animals (257,279 SNPs).

A density map of the single nucleotide variation in each chromosome is shown in Figure 1D,E, indicating a difference in the SNV density between the LZG and LBG breeds. The differences between the Y chromosomes are the most striking, where the SNV-free regions appear to be located at different coordinates, highlighting the differences in the paternal origins and confirming the unrelated origin of the two breeds. In addition, the extent of the SNV-free regions was greater in the LZG breed than in the LBG breed. According to the graphs, there are genomic regions with a very high (f1) density of SNPs on chromosomes 10, 12, and 23. Regions comprising a high nucleotide diversity correspond to the variation hotspots that are characteristic of these bovine chromosomes. For example, the variation hotspot located within the window between 18 Mb and 35 Mb on chromosome 23, apart from other protein-coding genes, includes a major histocompatibility complex locus [45,46]. In contrast to the low number of high SNV density regions (yellow to red), many regions displayed a low SNV density (grey to dark green) within the genomes of both breeds. In many cases, they overlapped; however, it was possible to visually identify several chromosomal regions that are unique to each breed.

### 3.2. Population Genetics

Comparing local and other widespread European breeds revealed considerable differences. The PCA analysis showed a pronounced genetic distinction between the LZG and LBG breeds (Figure 2A), as well as between the LBG and European breeds (Figure 2B). The highest variance was explained by PC1 (13.5% for LBG and LZG; 12.1% for LBG, LZG, and European breeds) and PC2 (7.2% for LBG and LZG; 7.93% for LBG, LZG, and European breeds), and the explained variance in the subsequent decrease in PCs led to the diffusion of sample clusters. Several animals representing the LZG breed were grouped close to the cluster formed by other European cattle breeds, which included Holstein. The observed clustering patterns agreed with the data on the ancestry of animals representing the two local breeds, with LZG animals sharing a higher proportion of Holstein ancestry.

A linkage disequilibrium (LD) analysis showed that the LZG breed had a slightly slower LD decay rate than the LBG breed (Figure 2C), but the two were highly similar. The highest LD decay rate was observed for HLS and TYG, and the lowest LD decay rate was observed for the LMS breed.

The estimated genome-wide nucleotide divergence (π) within a 50 kb sliding window ranged from 5.0 × 10^−7^ to 0.0323 (median: 1.44 × 10^−3^; mean: 1.79 × 10^−3^) in the LBG samples (Figure 2E and Appendix A) and from 5.0 × 10^−7^ to 0.0357 (median: 1.46 × 10^−3^; mean: 1.81 × 10^−3^) in the LZG samples (Figure 2E and Appendix A). The highest nucleotide diversity was observed in the autosomes, followed by the X and Y chromosomes. Overall, the nucleotide diversity results show a low genetic diversity that differs significantly between the breeds (*p* = 0.01249). Next, the LBG and LZG sample sets were randomly rarefied to three samples per breed to compare the nucleotide diversity between the other European breeds. The nucleotide diversity of LBG differed significantly from that of LMS (*p* = 0), LZG (*p* = 1.39 × 10^−10^), and HLS (*p* = 1.20 × 10^−135^), whereas LZG significantly differed from that of TYG (*p* = 1.51 × 10^−7^), BBL (*p* = 1.08 × 10^−18^), GER (*p* = 8.12 × 10^−7^), HLS (*p* = 3.56 × 10^−69^), HRF (*p* = 8.74 × 10^−7^), and LMS (*p* = 0) (Appendix A). Overall, the highest average nucleotide diversity was observed for the HRF breed (median: 1.443 × 10^−3^; mean: 1.81 × 10^−3^), followed by the BBL breed (median: 1.429 × 10^−3^; mean: 1.85 × 10^−3^). The lowest average nucleotide diversity was observed for the LMS breed (median: 1.06 × 10^−3^; mean: 1.38 × 10^−3^), followed by the HLS breed (median: 1.18 × 10^−3^; mean: 1.59 × 10^−3^). The level of heterozygosity (Figure 2F,G) was also generally low and did not differ significantly between the breeds (*p* = 0.057); the level of heterozygosity was 0.2437 ± 0.031 for the LBG breed and 0.2384 ± 0.019 for the LZG breed. When the randomly rarefied sample depths of the LBG and LZG breeds were compared to those of the European breeds, we uncovered that HLS and LMS (*p* = 0.016) and LMS and TYG (*p* = 0.049) differed significantly in terms of heterozygosity (Appendix A). The highest levels of heterozygosity were identified for TYG (median: 0.4881; mean: 0.4735) and HLS (median: 0.4626; mean: 0.4557), and the lowest levels were observed for LMS (median: 0.2827; mean: 0.2816) and HRF (median: 0.3377; mean: 0.3418). Notably, the rarefied sample sets of LZG and LBG showed higher values of heterozygous sites, whereas the full data set displayed considerably lower values, thus highlighting the need for large sample sizes when conducting WGS-based population research.

We also calculated Tajima’s D statistic to understand whether there was selective pressure on the population. Tajima’s D statistic differed significantly between the LZG and LBG breeds (*p* = 2.2 × 10^16^). The average Tajima’s D statistic across all chromosomes was 0.769 ± 1.14 for the LBG breed (min = −2.94, max = 4.47, and median = 0.78) (Appendix A), whereas for the LZG breed, it was 0.601 ± 1.14 (min = −2.94, max = 4.48, and median = 0.59) (Appendix A), indicating that there are few rare alleles. Positive Tajima’s D values (Tajima’s D > 0) were observed for the majority of the LZG (70.54%) and LBG (75.19%) genomes, while negative values (Tajima’s D < 0) accounted for approx. 24.81% of the LBG genome and 29.46% of the LZG genome, thus indicating the high abundance of rare alleles due to selective pressure and the theoretical population expansion. An excess of rare alleles was observed on most autosomes and both sex chromosomes, mostly at different locations on each chromosome. The lowest Tajima’s D statistic was observed on chromosomes 8, 17, 13, 5, 11, Y, 25, and 19 for the LBG breed and on chromosomes 7, 16, X, 1, 21, Y, 9, 2, and 14 for the LZG breed.

In total, 422,266 ROH were obtained from the LBG breed data set, from which 10 were long (≥1000 kb), and 185 were of medium length (1000–500 kb). From the LZG breed, 416,381 ROH were identified; 3 of these were long (≥1000 kb), while 97 were of medium length (1000–500 kb). The majority of identified ROH in both breeds were relatively short. The number of ROH between 50 kb and 500 kb in length was 128,594 and 121,818 for the LBG and LZG breeds, respectively (Figure 2D). The average number of ROHs longer than 50 kb covering autosomes ranged from 3150.3 ± 668.6 among the LBG animals to 2863.9 ± 589.5 among animals of the LZG breed. The median length of ROH across all chromosomes was 32.97 kb for the LBG breed and 32.32 kb for the LZG breed. Longer ROHs were observed on the X chromosome of the LBG breed compared with that of the LZG breed; however, European breeds, notably BBL, GER, and LMS, displayed the longest average ROH on the X chromosome.

To find the degree of differentiation between both local cattle breeds of Latvia, the F_ST_ statistics were calculated between individuals of the LZG and LBG breeds by each chromosome (Appendix A). The highest weighted F_ST_ statistics were observed for chr6:66.36~66.40 Mbp (F_ST_ = 0.72), chr11:18.60~18.64 Mbp (F_ST_ = 0.64), and chr6:61.94~61.98 Mbp (F_ST_ = 0.64). In contrast, the lowest weighted F_ST_ statistics were observed for chr1:158.02~158.06 Mbp (F_ST_ = −0.03) and multiple regions within the Y chromosome—9.58~9.62 Mbp; 11.06~11.12 Mbp; 11.96~12.02 Mbp; and 15.22~15.80 Mbp (for all F_ST_ = −0.03). Overall, the median of the weighted F_ST_ statistics across all autosomes was 0.047, whereas across both sex chromosomes, it was 0.037.

To assess the historical admixture patterns of the LZG and LBG breeds, we conducted an admixture analysis with K values ranging from 2 to 7 (Figure 3). At K = 2 and K = 3, the LBG and LZG breeds shared the same ancestral lines with major European *Bos taurus* breeds. Starting from K = 4, the LZG and LBG breeds were separated from traditional European breeds, indicating a further genetic differentiation process. According to the calculated cross-validation value, the best fit was K = 1 (CV error = 0.45504), meaning that all breeds included in the admixture analysis have one ancestral component that is different from others. This finding is in concordance with the PCA results (Figure 2B), showing that both local Latvian *Bos taurus* breeds are genetically distant from major European breeds.

In order to evaluate the genetic relatedness of the samples within the two sample groups (LZG and LBG), a genetic distance tree was constructed (see Appendix A). Overall, the genetic distance between the LBG and LZG animals was relatively low, with the proportion of different loci at ≈10%. Upon analysis of the LZG sample group, it was found that the animals of the SLZG_Turaida group, followed by those of the SLZG_Sidra, SLZG_Zilupe, and SLZG_Salna groups, were the most genetically distant among all of the LZG samples. Genetic relatedness was confirmed, for example, between the deep-sequenced LZB_Karlos and LZB_AL (son and father). Slight variations in the genetic relatedness patterns were revealed from the pedigree information available at AIC. Four related animals of the LZG breed grouped with the LBG animals. All four were descended from the same paternal lineage, and three were closely related (two half-sisters and a sister’s daughter). The LBG animals’ forming clusters and flanking groups of the LZG cows represent two major LBG paternal lineages. The use of sires from the Potrimps lineage for crossbreeding has been well documented, but there are no data on the admixture of animals from the LBG lineage Odins in the LZG breed. Since the overall genetic distances were low, the distribution of animals within the genetic tree could have been affected by an admixture of different breeds at different proportions within ancestral lineages. Nevertheless, the ancestry of local cattle has been documented in detail only with the beginning of artificial selection. One of the explanations could be that there is some proportion of shared ancestry between the two local cattle breeds. The pedigree data may be incomplete, and errors in the pedigree records cannot be excluded. For instance, although SLZG_Dalija and SLZG_Durbe were sisters, their genetic distance was considerably greater than expected. Similarly, the genetic distance between SLZG_Dalija and her daughter, SLZG_Dadze, was notable.

The inbreeding rate among the animals was assessed using genome-wide data on ROH. The F_ROH_ based on the sum of all ROH taken together was slightly but significantly higher in the LBG breed (0.197) than in the LZG breed (0.179; *p* = 0.038).

The inbreeding measures that were calculated considering only ROH that were longer than 50 kb or F_ROH>50kb_ reached 0.12 for the LBG breed and 0.10 for the LZG breed (*p* = 0.02). However, the respective standard deviations of 0.039 and 0.033 for F_ROH_ and F_ROH>50kb_ in the LBG breed and 0.029 and 0.027 in the case of the LZG breed indicated a relatively high variation in interrelatedness among individual animals from both breeds.

### 3.3. Variant Association with Mastitis

The level of SCC did not differ significantly (*p* > 0.05) between the LBG and LZG cow breeds in the first lactation, at the time of pathogen testing, and in the highest SCC scores in later lactations (Figure 4A). During pathogen testing, the SCC levels were higher (*p* < 0.05) than in the first lactation in the LBG cows but not in the LZG cows. An overall comparison of the average SCC between farms was nonsignificant (*p* > 0.05). There was a moderate correlation between the SCC during the first lactation and the SCC during bacteriological testing (*r* = 0.418; *p* < 0.05).

Bacterial growth was confirmed in milk samples of 16 LBG animals, but no bacterial growth was detected in the other 16. The milk of only 11 LZG animals was confirmed to contain bacteria; there were none in the samples from 10 animals. The increase in SCC coincided with the number of major pathogens detected in the milk. From the tested set of bacterial species, six (*Enterococcus faecalis*, *Lactococcus lactis*, and the *Streptococcus* species *S. uberis*, *S. xylosus*, *S. dysgalactiae*, and *S.simulans*) were found only in the LBG milk samples, while three (*Aerococcus viridans*, *Streptococcus mitis*, and *Staphylococcus simulans*) were specific to the LZG samples. Variations in the prevalence of bacteria in milk samples could be due to environmental factors and differences in resistance toward pathogens between the two breeds. The prevalence of bacterial agents in the cow milk samples is shown in Figure 4B.

Taking both breeds together, 15.63% of the milk samples showed no bacterial growth. An analysis of the bacterial taxa within positive samples determined that the dominant species was *S. haemolyticus* (17.19%), followed by other Gram-positive bacteria *Staphylococcus epidermidis* and *S. aureus* (6.3% each), as well as *E. faecalis* and *S. uberis* (4.7% each). Therefore, during the subsequent genome-wide association analysis, we considered the consanguinity of the animals and the results that were gained during the microbiological analysis of the milk. Thus, the genomic data from 16 healthy and 16 mastitic LBG breed animals, as well as 10 healthy and 11 mastitic LZG breed animals were used to investigate genetic traits linked to mastitis resistance (Appendix A). Four LBG cows and fifteen LZG cows were undiagnosed, because at the time of sample collection, they were either dry cows or deceased. The acquired genome-wide association results revealed one LBG breed variant (Figure 4C) and three LZG breed variants (Figure 4F) associated with a resistance to mastitis pathogens that passed the significance threshold of *p* < 1.0 × 10^−9^ and the Bonferroni-corrected *p*-value of <0.05.

In the case of the LBG breed, the quantile–quantile (Q-Q) plot (Figure 4D) indicated a well-controlled population stratification, as evidenced by the slight deviation of the observed distribution of *p*-values from the expected distribution and by the genomic inflation factor lambda (λ = 1.23). Thus, the identified mastitis-associated SNP rs721295390T > A was located in the intergenic region on chromosome 10 (OR = 0.019, *p* = 0.0254). Only 2.6 kb upstream of this SNP was the location of the reverse-orientated *LOC529823* gene that encodes C2 calcium-dependent domain-containing protein 4A, while further on in the distance of 19.8 kb was the forward-orientated gene *VPS13C* that encodes vacuolar protein sorting 13 homolog C. The frequency of this SNP in mastitis-affected cows was 0.09, whereas in healthy cows, it was 0.844. Furthermore, 14 diseased, 2 healthy, and 4 undiagnosed cows were homozygous for the reference allele (T/T), whereas 13 healthy cows and 1 diseased cow were homozygous for the alternative allele (A/A). Only one diseased, one healthy, and four undiagnosed cows were heterozygous (T/A) at the particular SNP location.

For the LZG breed, in the Q-Q plot (Figure 4G), a deviation in the upper right tail from the null distribution confirmed evidence of a weak association and population stratification (λ = 1.74), which was most likely due to the low number of animals included in the analysis. As a consequence, it is advised to approach the subsequent findings with caution, considering them as suggestive rather than conclusive outcomes. Thus, mastitis-associated SNPs (rs383806754, chr29:43998719CG>C, and rs462030680) were located on chromosomes 17, 22, and 29. Two of them—rs383806754 and chr29:43998719CG>C—were located within the intron of calcium-dependent secretion activator (*CADPS*) on chromosome 22 and cathepsin W (*CTSW*) on chromosome 29, respectively, and were accommodated between genes involved in the regulation of the immune system. Thus, the latter chr29:43998719CG>C (*p* = 0.003) was located 784 bases downstream of the treverse-oriented gene *FIBP* that encodes FGF1 intracellular binding protein and 6.6 kb upstream of the reverse-oriented gene *EGF* that encodes fibulin extracellular matrix protein 2 (*EFEMP2)*. Both encoded proteins play a role in the regulation of inflammatory processes. The same SNP chr29:43998719CG>C was also located 6.9 kb downstream of the forward-oriented gene *CCDC85B*, which encodes a coiled-coil domain-containing protein 85B. It functions as a transcriptional repressor involved in maintaining the cytoskeleton integrity of epithelial cells. The frequency of this SNP chr29:43998719CG>C in mastitis-affected cows was 1, whereas in healthy cows, it was 0. More specifically, 11 diseased and 6 undiagnosed cows were homozygous to the reference allele (CG/CG), whereas 9 healthy and 10 undiagnosed cows were homozygous to the alternate allele (C/C). Only one healthy and three undiagnosed cows were heterozygous (CG/C). The other SNP in close proximity to the immune-system-related genes was rs462030680 (*p* = 0.009), located 134 bases upstream of the reverse-oriented gene *FBRSL1* that encodes fibrosin-like 1 protein, which is suggested to be essential for stem cell maintenance and differentiation in mammals. The frequency of this SNP in mastitis-affected cows was 0.05, and in healthy cows, it was 1. In total, 10 diseased and 11 undiagnosed cows were homozygous for the reference allele (C/C), 10 healthy and 8 undiagnosed cows were homozygous for the alternative allele (G/G), and only 1 mastitis-diseased cow was heterozygous for (C/G). The last of the identified SNPs, rs383806754 (*p* = 0.001), which was mentioned previously, located within the intron of the *CADPS* gene, which encodes calcium-dependent secretion activator, was found exclusively in mastitis-diseased cows (allele frequency 1), and it was absent in the healthy controls (allele frequency 0). More specifically, 11 of the diseased cows and 8 of the undiagnosed cows were homozygous for the reference allele (TG/TG), while 10 healthy and 10 undiagnosed cows were homozygous for the alternate allele (T/T), and only 1 undiagnosed cow was heterozygous (TG/T).

## 4. Discussion

The increased popularity of highly productive commercial cattle breeds has led to a dramatic decline in local dairy cattle breeds and the loss of valuable genetic resources. Two local Latvian breeds have small population sizes and are critically endangered. Knowledge of the genetic structure of the breed can provide essential support to improve breeding programs and to preserve the genetic integrity of local cattle breeds, which inspired us to perform whole genome sequencing analysis.

### 4.1. Genetic Structure of Breeds

The characterization of relatedness between the LBG and LZG breeds showed that both had retained at least some breed-specific genetic hallmarks. Although the differences were minor, and the admixture analysis indicated shared origins with other European breeds, the clustering patterns suggested distinct ancestry. Historically, LBG cattle have been crossbred with Traditional Danish Red and German Red Angler cattle, whereas LZG cattle had a stronger influence from Holstein cattle and several other breeds. Thus, an admixture of other European cattle might have disrupted breed-specific patterns. Apart from that, using different breeds to improve the LBG and LZG cattle may have contributed to distinct clustering patterns of both breeds observed in both the PCA and admixture analysis. This difference in admixture source can also explain the slightly larger distance between the LBG cluster and other breeds included in PCA. According to the genome-wide data, the Modern Danish Red, German Red Angler, and Modern Angler cattle appeared to have common ancestry, distinct from the closely related Holstein and Holstein Red breeds [46]. Previous marker-based genetic studies investigating cattle breeds native to northern Europe and Baltic countries have firmly placed the LBG breed among other Red cattle breeds, which are closely related to the Danish Red breed and include Lithuanian and Estonian Red cattle. In contrast, according to microsatellite data, the ancestry of the LZG breed was less clear because it was shown to be closely related to LBG and other Red cattle breeds, yet it also had a quite different genetic background [47,48]. These discrepancies between studies could most likely be explained by different crossbreeding rates with other breeds, including LBG. Nevertheless, both studies agree on a highly mixed breed composition of LZG, which also showed a higher level of clustering in the current analysis. These observations correspond with pedigree-derived data on the purity of ancestry of individual animals included in our study [47,48]. The situation with LZG is similar to that described for Estonian Grey cattle, where the original native breed has often been crossbred, particularly with Holstein [48].

Regarding the F_ST_ values (median value of F_ST_ 0.05), there was relatively low genetic differentiation between the two local breeds, and the short genetic distances among the LZG and LBG animals shown via dendrogram indicated a close interaction. For instance, the endangered breed German Black Pied cattle had an F_ST_ value of 0.069 compared to Holstein, supporting a shared history, whereas a three-times-higher value of F_ST_ 0.175 was estimated for Yakutian cattle, an old, genetically distinct native breed from Siberia with no influence from modern European cattle [46]. Correlation patterns among individual animals based on genetic distances were close to those known from pedigrees, though not always providing a direct match. However, there is a high rate of relatedness among animals representing each breed, shared ancestry between LZG and LBG animals (e.g., LZG sire Samts was from one of the major LBG paternal lineages Potrimps), and to no lesser degree, variable rates of admixture from several other breeds have affected constructed relationship patterns. Hierarchically, the most distant animal of the LZG breed, Turaida, was probably placed at the root of the genetic tree because of the kinship with major paternal ancestral lineages of LZG cattle—the sub-lineages Jumis and Aizups of the Gaujars Lietuvietis lineage and LBG cattle of the Rudme and Potrimps lineages. Two dams, half-siblings sired by the same purebred Lithuanian Light Grey bull, formed a separate cluster from other animals of Latvian breeds, which fits well with previous microsatellite marker data. Both breeds also differ by phenotype, strongly suggesting different histories for Latvian and Lithuanian grey-coated cattle [47]. None of the animals included in the current genome-wide analysis had ancestry from Estonian Gray cattle, but according to microsatellite data, the two breeds had distinct origins. Holstein ancestry among studied Estonian animals may have obscured genetic patterns that are characteristic of the original breed [48].

The total average nucleotide diversity estimated for LBG and LZG indicated that Latvian breeds harbor relatively high levels of genomic variation compared with Holstein cattle. In our case, subsampling data from publicly available resources may have biased the estimates for other breeds, which appears to be the case for Limousin cattle [46]. However, we believe that our findings are supported by research conducted by other authors. The genetic diversity of Holstein cattle has been depleted by long-term intensive selection [49]. Similarly, other native locally distributed breeds such as East and West Finn cattle, Original Braunvieh, German Black Pied cattle, and Yakutian cattle have been shown to have higher genetic diversity than mainstream breeds [46,50,51].

Heterozygosity is another measure that is often used to track the amount of genetic variation retained by the population. It was higher among the LBG animals than the LZG animals, which is opposite to the estimated nucleotide diversity that was slightly lower for the LBG breed. This result was probably due to the high proportion of offspring of a single LZG bull. Similarly, the discrepancy between the two estimates among German Black Pied cattle, a high rate of nucleotide diversity, and low heterozygosity was explained by the limited number of breeding bulls [46]. In general, the heterozygosity estimates for the local cattle were lower than those observed among other European cattle breeds, including Holstein, and were closer to those observed in breeds characterized by high inbreeding levels, such as Jersey cattle [46,52,53,54]. However, the majority of available data have been generated by standard SNP genotyping assays that can yield higher heterozygosity measures than those derived from WGS, which hampers direct comparison [55].

Genome-wide LD-based measures serve as good indicators of the processes shaping a breed’s genetic background [56]. A moderate, gradual decrease in LD observed for the two local breeds was close to those observed for the German Angus, Belgian Blue, and Hereford cattle breeds, which ranged between the results obtained for the Holstein and Limousin cattle breeds. The Holstein breed is well known to have a high proportion of genome covered by ROH accompanied by slow LD decay as a consequence of intensive selection [49]. Limousin cattle, on the other hand, have been repeatedly shown to harbor higher genomic diversity and have comparatively low levels of inbreeding [46,53,57].

A relatively slow LD decay, moderate levels of heterozygosity, and measures based on the proportion of genome covered by ROH indicate comparatively high inbreeding for the two local breeds. Among Alpine cattle, the highest inbreeding rate F_ROH_ of 0.088, estimated by summing all the ROH lengths per individual, was observed for the Evolene breed. The total contemporary population of these animals is around 200 cows. Despite a similar population size, the inbreeding rate of Evolene cattle was notably lower than F_ROH_ 0.197 and 0.179 observed in the LBG and LZG breeds, respectively [52]. On the other hand, while local cattle had F_ROH>50kb_ values of 0.12 and 0.10, Neumann and colleagues reported a slightly higher inbreeding coefficient of 0.16 for German Black Pied cattle native to the North Sea region, derived from the sum of ROH longer than 50 kb. The breed is endangered, and its population consists of approximately 2500 animals. Although the F_ROH>50kb_ values of other breeds that were included in the study varied from 0.14 for Modern Danish Red to 0.29 for Jersey, which typically indicates increased inbreeding, for the Hereford, Angus, and Holstein breeds, the F_ROH>50kb_ values were 0.26, 0.23, and 0.20, respectively [46].

Our data were more comparable with the inbreeding rates obtained by studies investigating the genetic diversity of Original Braunvieh cattle and four Danish dairy cattle breeds with different demographic histories [50,58]. Among the Danish dairy cattle, the inbreeding levels varied between 0.12 for New Red Danish cattle and 0.24 for Jersey. The Holstein cattle showed ROH-based inbreeding levels of 0.19, which matched with the F_ROH_ value in the LBG breed [58]. Original Braunvieh is a dual-purpose cattle breed adapted to alpine pastures. It has recently experienced a bottleneck, and the average genomic F_ROH>50kb_ value for these animals was 0.14 [50]. This result is slightly higher than that derived from the genome-wide data in the LBG (0.12) and LZG (0.10) breeds. However, the average length of ROH estimated for the LBG and LZG breeds in our study was lower than generally observed among other cattle breeds, which might have resulted in the underestimation of inbreeding measures.

According to an analysis of blood group alleles in Baltic cattle breeds, LZG showed the lowest diversity of alleles. The homozygosity levels for LBG cattle were missing, but the degrees of homozygosity in the Estonian Red and Lithuanian Red cattle breeds were 5.4% and 7.4% [59]. Pedigree-based inbreeding coefficients likewise support a high rate of consanguinity among animals, particularly of the LZG breed. Over the past decade, a steady increase in inbreeding was observed within Latvian breeds, reaching 2.61% for LBG and 5.20% for LZG breeds by 2019 [60].

Although the blood group typing data and pedigree-derived measures of relatedness (F_PED_) observed by others and the inbreeding rate estimated from genome-wide data were in agreement on both populations being inbred, inbreeding coefficients based on ROH were at least twice as high and contradictory to F_PED_ data, were increased in LZG compared with LBG cattle [59,60]. The pedigree-based assessment of inbreeding was limited by the completeness and reliability of the pedigree data, and it is only a proxy indicating the proportion of the genome expected to be identical by descent [61]. The pedigree completeness for the LBG and LZG animals included in the inbreeding evaluation study was very good [60]. However, the parentage records might have been less reliable for the LZG breed. The breeding of LZG animals is less centralized, with small numbers of animals dispersed across numerous dairy farms, reducing the traceability of actual animals’ parentage. However, the discrepancy in inbreeding rates is more likely explained by the demographic history of breeds. The LBG breed is a more homogenous breed than LZG. Animals of the LBG breed had, on average, considerably more ROH that were longer than 500 kb covering genome compared with the LZG animals, which could have affected the F_ROH_ estimates. The structure of the ROH in the genomes of the LZG cattle could be disrupted by a high rate of crossbreeding with several different breeds. Similarly, among sheep breeds, genomes of crossbreed animals comprised significantly less ROH than purebred ones [62].

### 4.2. Mastitis Resistance

The number of animals available for the conservation of local LBG and LZG breeds is minimal. Knowledge of genetic risk factors associated with diseases like mastitis can guide the selection of animals for breeding to improve disease resistance. Therefore, we performed an association analysis to detect genetic traits associated with the development of mastitis.

The Q-Q plot and genomic inflation estimator lambda were used in this work to assess the effectiveness of the GWAS models. For the LBG breed, both the Q-Q plot and lambda measures revealed evidence of a weak population stratification, which is a finding that is consistent with recent studies [63]. This suggests that the sample size of mastitic and healthy cows within the LBG breed was nearly adequate to obtain reliable results on significant SNPs associated with the condition. On the other hand, when analyzing the LZG breed, the Q-Q plot and lambda value indicated a weak association and raised concerns about potential population stratification. This finding mirrors observations from previous whole-genome population studies of animals [64]. One critical aspect that emerged from the quality measures was the relatively low number of animals included in the association analysis. This limitation might have influenced the statistical power and precision of our findings. However, it is worth noting that similar studies with a comparable number of animals have previously been conducted [63,64,65,66,67,68]. Despite this constraint, the results and knowledge obtained offer valuable insights and could serve as a foundation for future investigations related to Latvian local cattle breeds.

In this study, we found four genetic variants associated with mastitis among LBG and LZG cattle. Two were located within intergenic regions, whereas the other two were found within the intronic regions, e.g., one in the *CTSW*, and the other in the *CADPS* gene. In the majority of cases, the regulation of genes closest to the association signal is affected by the identified genetic variation, providing that the signal comes from the causal variant and not a proxy [69]. In line with findings from recent meta-analyses, association signals detected among LZG and LBG animals were located within proximity of genes with plausible roles in the development of mastitis [15]. The closest gene to the single variant that correlated with the prevalence of mastitis among LBG cows was *C2CD4A*, which encodes a transcription factor that is widely known for its role in the development of diabetes [70]. Although the role of *C2CD4A* in insulin secretion could theoretically be linked with milk-production-related traits, *C2CD4A* most likely contributes to the development of mastitis through its involvement in inflammation-related processes [70]. The transcription factor *C2CD4A* regulates genes that modify the architecture and adhesion of epithelial cells in response to the pro-inflammatory cytokines as a part of the NF-κB signaling cascade [71]. Further upstream of the target SNV is the location of *VPS13C*, which has been previously placed among genes that are differentially expressed in response to the *E. coli* infection of bovine mammary gland epithelial cells [72].

Genetic variants located within the genomic region on chromosome 29 overlapping *CTSW* and the two other genes found within the neighborhood, *FIBP* and *EFEMP2*, have been reported among quantitative trait loci associated with the somatic cell count in Jersey cattle [19,73]. The causal link between the genes representing the locus and mastitis-related traits has not yet been established. The product of *CTSW*, Cathepsin W, is only expressed in cytotoxic T-cells and natural killer cells, where it is involved in regulating target cell killing [74]. The direct role in cell-mediated cytotoxicity makes *CTSW* a likely candidate involved in the pathophysiology of mastitis. The second-best candidate is probably *FIBP*, and it is also the nearest to the target SNVs. The FIBP binds fibroblast growth factor 1 (FGF1), which is involved in defense mechanisms against bacterial invasion in mastitis. The *FGF1* was among the genes induced by the *E. coli* infection of primary mammary epithelial cells in the udder [75]. The expression of *FGF1* induces calcium mobilization and the activation of NF-κB and other signaling pathways within immune cells with the net result of reduced inflammation [75,76]. The role that *EFEMP2* could play in the development of mastitis is less clear. The results of previous GWAS studies suggested that the *EFEMP2* locus is affecting cattle meat quality [77]. Another upstream-located gene that encoded transcriptional repressor *CCDC85B* may promote the progress of inflammation of udder tissue through mechanisms related to those of *C2CD4A*. The activation of *CCDC85B* enhances β-catenin degradation, retaining NF-κB activity and thus inhibiting the Wnt/β-catenin cascade. The inhibition of the Wnt/β-catenin pathway prevents the re-organization of the cytoskeleton of epithelial cells. Preserving the cytoskeleton structure maintains the integrity of the endothelial barrier and impedes bacterial invasion within the tissue [77,78,79]. Another association signal for mastitis among LZG cows was located within the intron of *CADPS*. The locus harbors genetic variants correlating with body conformation traits, including bone quality, leg and feet traits, and eating behavior among cattle [80]. Other studies have linked the locus with the reproductive traits of cattle [81]. In humans, the increased expression of *CADPS* within immune cells has been shown to modulate inflammatory responses of intestinal endothelium during chronic inflammation, which is characteristic of celiac disease [82]. It can be speculated that *CADPS* may be involved in the development of mastitis through multiple mechanisms [80,81,82]. Yet, *FEZF2* found upstream from the rs383806754 has been linked with mastitis in other studies [20,83,84]. A copy number variant located within the gene increased the susceptibility to mastitis by almost two times among cattle in a study conducted by Japanese researchers. Furthermore, *FEZF2* expression was increased in the mastitic quarters of affected cows [20,84]. The *FEZF2* was suggested to be involved in immune tolerance and defects of the gene, causing autoimmune diseases in humans [85]. It cannot be excluded that variants located within the intronic region of *CADPS* may have a regulatory effect on the expression of *FEZF2*.

The fourth locus that correlates with mastitis among LZG animals comprises *FBRSL1,* which encodes a protein with RNA binding activity belonging to the polycomb complex [86]. In humans, a homolog of this gene was recently connected with the severity of COVID-19 disease, suggesting a possible link with immune response [87]. Its functions in cattle have not been studied in detail, but several studies have demonstrated the correlation between variants within the locus and milk yield, udder development, and udder depth [88,89]. Udder conformation, particularly the structure of the teat channel, significantly affects microbial invasion and the subsequent inflammatory response [90].

To sum up, the acquired SNP profiles were distinct for each breed, suggesting that the genetic composition of the LZG and LBG breeds influences the association signal. Although genetics appears to have a significant influence on the association analysis outcomes, it is important to consider the potential impact of environmental factors on mastitis development. Previous studies have highlighted the contribution of various factors, such as cattle housing conditions and milking routines, among others, to the manifestation of mastitis [11,12,13,14,17,18]. Consequently, it is highly plausible that mastitis-associated SNP profiles in each breed are the result of a combination of genetic and environmental factors, which warrants further detailed investigation in the near future.

The conducted association analysis supports the presence of association signals within previously detected candidate genomic regions on chromosomes [19,73,83,84] and provides insight into the genetic background of mastitis resistance in local cattle populations. Yet, the contribution of potential risk variants requires further verification. The major limitation of this study is the low number of animals included in the association analysis. Despite efforts to mitigate erroneous signals, the small sample size, particularly in the LZG breed with a heterogeneous genetic background, may have led to false positive findings or masked genuine associations. Furthermore, as the animals used in the analysis originated from different herds, the impact of uncontrolled environmental confounders could not be entirely ruled out. To strengthen the validity of the results, the currently obtained association signals should be replicated in a larger number of animals representing both breeds, preferably in larger cattle populations. Furthermore, before concluding the significance of the candidate loci in mastitis resistance, a causal link must be established. Given the intricate nature of genetic resistance to bovine mastitis, the current findings nevertheless provide valuable guidance for further investigations into causal genetic risk factors, particularly in these two local cattle breeds.

## 5. Conclusions

We completed a genome-wide characterization of two indigenous cattle breeds of Latvia. Even though the crossbreeding of LZG cattle is particularly common, the animals representing the LBG and LZG breeds still carry some distinctive genetic hallmarks that are characteristic of the original ancestral breeds. Until now, LBG and LZG cattle breeds have retained substantial genetic diversity. However, moderate levels of heterozygosity and elevated inbreeding indices indicate an urgent need to preserve both breeds’ genetic integrity. Some proportion of original genetic traits may have been already irreversibly lost, which hinders the restoration of the original genetic profile of both breeds. However, local breeds are still of genetic value and represent a part of cultural heritage. To preserve these resources, it would be necessary to continue the genetic characterization of the remaining cattle. The accumulation of genetic data would facilitate the selection of the most representative animals for breeding and help to choose breeds that are most suitable for crossbreeding, which might be necessary to retain healthy genetic diversity. Highlighting genetic variants that potentially increase disease susceptibility can improve pedigree-based breeding programs by reducing the possibility of the accumulation of multiple risk alleles that compromise the fitness of LBG and LZG breeds. Genomic data could also be used to establish ‘living gene banks’, improving the odds of preserving local cattle breeds. Apart from assisting in the management of national breeds, research on local breeds improves our knowledge of genetic resources that are globally available to fulfill the requirements of cattle breeding development.

## Figures and Tables

**Figure 1 animals-13-02776-f001:**
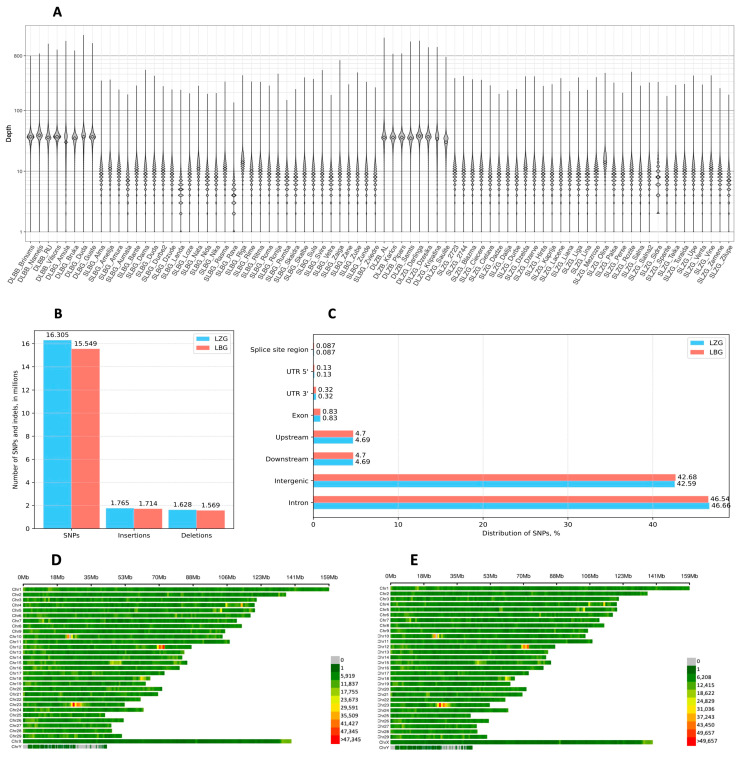
Sequencing depth of called variants and their distribution. (**A**) Sequencing depth of called variants for the LZG and LBG sample set. DLZB (bulls of LZG set), DLZG (cows of LZG set), DLBB (bulls of LBG set), and DLBG (cows of LBG set) samples were sequenced with deep target coverage, while samples from SLZG (cows of LZG) and SLBG (cows of LBG) groups were sequenced with shallow target coverage. (**B**) Distribution of variants among LZG and LBG breeds. (**C**) Distribution of SNPs in LZG and LBG breeds. (**D**) The density of SNVs within a 1 Mb window size for each chromosome of LZG breed animals. (**E**) The density of SNVs within a 1 Mb window size for each chromosome of LBG breed animals.

**Figure 2 animals-13-02776-f002:**
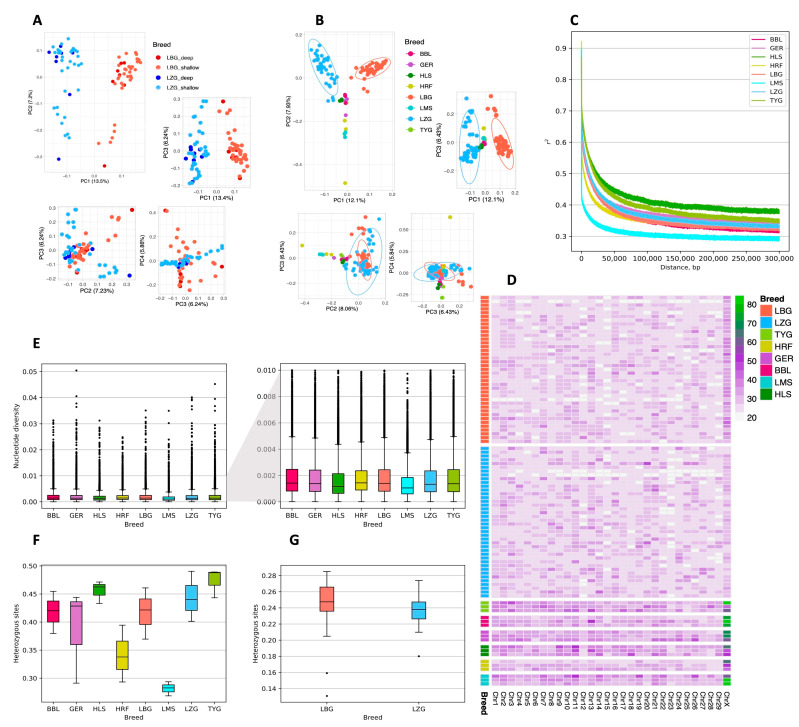
Genome diversity and linkage disequilibrium (LD) decay per breed. (**A**) Principal component analysis (PCA) of LBG and LZG breeds. (**B**) PCA analysis of all breeds included in the study. (**C**) LD decay for each breed. LBG and LZG sample sets were randomly reduced to 3 samples per breed. (**D**) Average runs of homozygosity (ROH) per chromosome in each breed. Rows represent samples. The green color represents longer ROH in kilobases, and purple represents shorter ROH. The length of ROH is presented in kilobases. (**E**) Genome-wide nucleotide diversity for all breeds. LBG and LZG sample sets were randomly reduced to 3 samples per breed. (**F**) Genome heterozygosity for all breeds. LBG and LZG sample sets were randomly reduced to 3 samples per breed. (**G**) Genome heterozygosity for a full sample set of LBG and LZG. Abbreviations: BBL—Belgian Blue; GER—German Angus; HLS—Holstein; HRF—Hereford; LBG—Latvian Brown; LMS—Limousin; LZG—Latvian Blue; TYG—Tyrolean Gray.

**Figure 3 animals-13-02776-f003:**
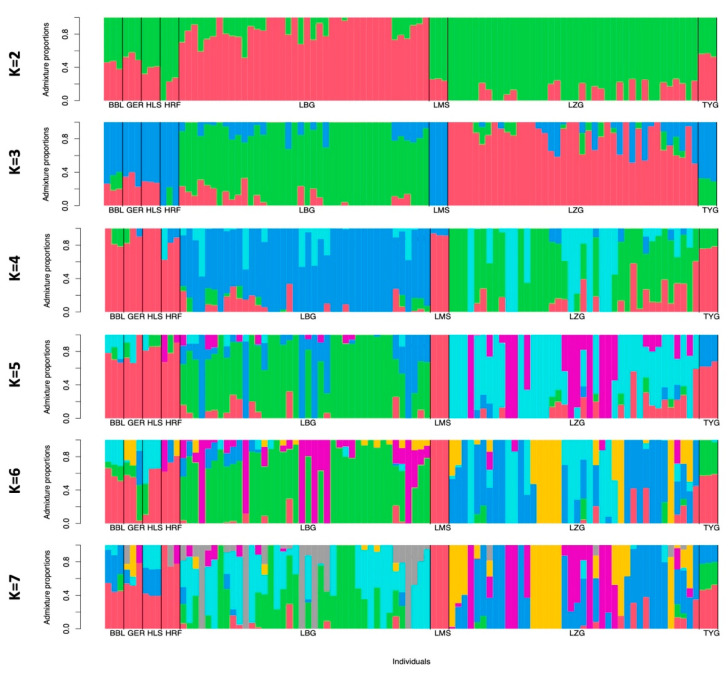
Historical population structure. Admixture analysis with K values ranging from 2 to 7. Each animal is represented by a single vertical line divided into K colors, where K is the number of clusters assumed, and the colored segment shows the individual’s estimated proportion of membership in that cluster. Black lines separate the populations labeled below the figure. Abbreviations: BBL—Belgian Blue; GER—German Angus; HLS—Holstein; HRF—Hereford; LBG—Latvian Brown; LMS—Limousin; LZG—Latvian Blue; TYG—Tyrolean Gray.

**Figure 4 animals-13-02776-f004:**
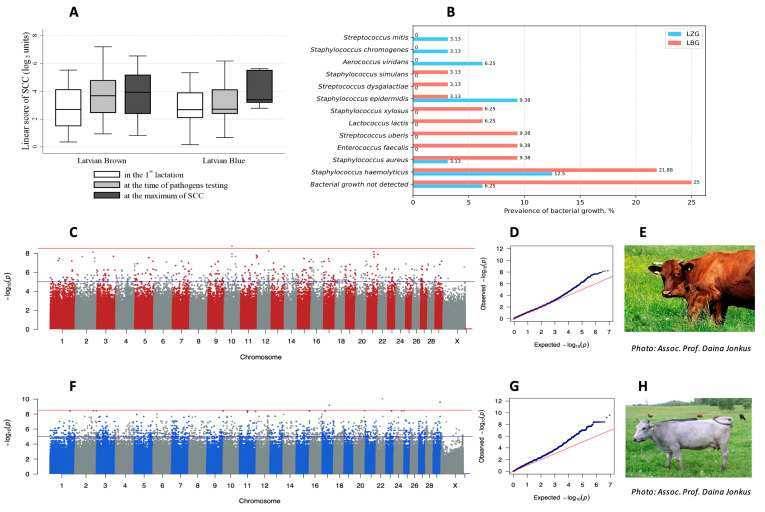
Milk analysis and Manhattan plot of genome-wide association with mastitis in local Latvian *Bos taurus* breeds. (**A**) SCC in the first lactation, at the time of pathogen testing, and at the maximum of SCC. (**B**) Prevalence (%) of bacterial pathogens in cow milk samples. (**C**) Genome-wide association with mastitis in LBG breed. (**D**) Q-Q plot of the association analysis of the LBG breed. (**E**) Photo of Latvian Brown cow breed in the pasture. (**F**) Genome-wide association with mastitis in the LZG breed. (**G**) Q-Q plot of the association analysis of the LZG breed. (**H**) Photo of Latvian Blue cow breeds in the pasture. Note: Manhattan plots display all nominal *p*-values from the association analysis for mastitis in breeds based on chromosomal position. In Manhattan plots, the blue line represents the genome-wide suggestive *p*-value threshold of −log10(1 × 10^−5^), whereas the red line is the genome-wide significant *p*-value threshold of −log10(3 × 10^−9^). In both Q-Q plots, the red line represents the expected *p*-value, whereas the blue dots represent the observed *p*-value from the association analysis.

## Data Availability

Raw sequencing data were deposited at the European Nucleotide Archive under study accession no. PRJEB60345.

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
