# Peer review of "Genomic Characterization and Initial Insight into Mastitis-Associated SNP Profiles of Local Latvian Bos taurus Breeds"

_animals, 2023, doi:10.3390/ani13172776_

Round 1
Reviewer 1 Report (Previous Reviewer 2)
My previous question: Have you checked the health management practices in the herd where you collected samples? The hygienic conditions are more or less the same in all herds?
I have not found the answer (Author’s previous reply) you provided before. I think it is important to add all this information in the manuscript.
My previous question: I think that in this paper is more correct to talk about mastitis resistance rather than mastitis susceptibility.
Ok for the changes but please check “susceptibility” at lines 108, 109, 823, and 876 and verify if this word is correct.
My previous question: I’m not sure that the term “breed” in the DAI-IS FAO database represents exactly a breed. Moreover, searching this database (Browse by country and species | Domestic Animal Diversity Information System (DAD-IS) | Food and Agriculture Organization of the United Nations (fao.org) - Latvia, Cattle) only LBG is considered at risk. Please consider this evidence in your paper.
I understood that LZG is considered endangered in Latvia but not for DAD-IS FAO. Please wrote and comment in the manuscript about this difference. If necessary, please change the information included in the FAO database because is the unique accessible for non-Latvian people.
Please check again the term “breed” you used.
My previous question: Have you read other papers using a very limited number of (related) samples in small populations? If yes please add to the text. If not, you must support the number of animals you used. In my opinion, the number of samples is too limited so, if you want to convince me you must write very well some things in support of your work. This is the major criticism of your work.
“Comparison with prior studies: In the Discussion section, we have now references several studies that conducted association analysis with comparable numbers of animals - Line:726-771. This addition provides a relevant context to the significance of our study's sample size.”
Please detail which studies you reported in the new version of the manuscript.
“Inclusion of Q-Q plots and lambda values: We have included Q-Q plots for both the LBG and LZG breeds in Figure 4. Additionally, the values of the genomic inflation estimator lambda for the LBG breed are now presented in Line:553-556, and for the LZG breed in Line:567-569.”
In my opinion, the lambda values are too far from 1.
My previous question: Considering your statement reported above you must consider for your work only the most significant markers, excluding the suggestive ones.
Answer?
My previous question: The proportion of variance explained by PC1 and PC2 is not very high (20.7% in total). Probably the inclusion of PC3 could be relevant for discriminating between breeds/populations.
Ok for your answer. But have you tried to prepare a graph including the three first PCs?
My previous question: The differences you found, in your opinion, are mainly due to environmental differences or to a genetic specificity of the two breeds? Please write a comment in the manuscript.
Your answer is not in agreement with what you wrote about the “environmental factors” = the health management practices. Please compare the two parts of the manuscript.
My previous question: CTSW and CADPS genes are overlapping? You stated that one genetic variant was found within the intronic region of two genes.
CTSW is located on cattle chromosome 29 while CADSP is on cattle chromosome 22. Please clarify the position of the two genes and of the detected SNPs in the manuscript both in the results and discussion sections.
none
Author Response
Reviewer 1
Please find below our answers to your comments point by point.

Reviewer 2 Report (Previous Reviewer 1)
Authors state in their response that:
We have summarized information on a number of herds and cows from each herd included in the study in the table provided in Supplementary data (Table S1, Sheet2)
In reviewing Table S1, I only find 1 sheet - Sheet1. Was Sheet2 omitted from the resubmission? Should there be another Table with herd information? It is not clear if herd information is included on Sheet1, but SCC data are evident.
I have no further comments
Author Response
Reviewer 2
Please find below our answers to your comments.

This manuscript is a resubmission of an earlier submission. The following is a list of the peer review reports and author responses from that submission.
Round 1
Reviewer 1 Report
This manuscript investigates the genome qualities of two breeds of Latvian cattle. The investigation and its presentation is extensive. The manuscript has two components, with the second related to mastitis traits.
Line 94 - “Exterior and meat quality”. It’s unclear what is meant by exterior. Does this refer to the environment or an additional meat or muscling trait? Clarify the term exterior or use a more definitive term for this trait.
Line 248 - extra ‘f’ in VCFtools.
Line 412 - missing ‘G’ on LZG
Lines 470-471 - although it is slightly higher, are these values really different?
Lines 484-491 - Here and related to mastitis in general: there is a strong effect of environment and management. The numbers of animals with culturable bacteria could be related purely to management and less so with genetic traits. The description of the number of herds used for sampling is difficult to follow in the methods. Can a definitive statement be made in the methods about the number of sites and how many animals were of each breed from each site?
Also, can a clarification be included about the animals with cultured bacteria and somatic cell counts be indicated by number of sites or herds? - were all the mastitic cows from one herd or multiple herds?
Line 499 - figure 4E
Line 511 - G to LZ
Line 526 - alternate or reference?
Can a forward looking statement about the potential to preserve or conserve these breeds be added to the Conclusions? Authors indicate a need to do so, but what is the potential for success given the inbreeding, limited stock, and crossbreeding likely to be happening. Is it realistic to believe these breeds will be preserved and what will be required?
Author Response
Manuscript ID: animals-2433556
Response to Reviewers
Dear Jessica Pan,
Thank you for giving us the opportunity to submit a revised version of the manuscript "Genomic characterization of two Latvian local Bos taurus breeds and identification of genetic traits associated with mastitis" for publication in the journal Animals. We appreciate You and the Reviewers for taking the necessary time and effort to review the manuscript. We are grateful for the valuable and constructive comments. These have greatly helped us to improve the quality of the current version of the manuscript. We have carefully considered the comments and incorporated necessary changes in the manuscript. Revisions are highlighted in yellow font within the manuscript. Please find below our point-by-point response to the reviewers' comments. Line numbers correspond to the revised version of the text. We hope the manuscript after careful revisions meets the expected standards to be suitable for publication in the journal Animals.
Reviewers' Comments to the Authors:
Reviewer 1
1) Comment - Line 94: “Exterior and meat quality”. It’s unclear what is meant by exterior. Does this refer to the environment or an additional meat or muscling trait? Clarify the term exterior or use a more definitive term for this trait.
Author response: We agree that use of the term "exterior" in the given context fails to convey the intended meaning of the sentence. Therefore, we have provided more detailed definition of traits linked with candidate genes related to mastitis which were previously summed under the term "exterior".
The sentence in line 94 "Others are associated with exterior and meat quality, reproduction, milk production-related traits, and health [15-17]." has been substituted with the following text:
Line 95-99: "Others are associated with milk production-related traits including milk yield, and composition of milk, as well as reproduction-related traits [15-17]. Some candidate genes have been linked with body conformation traits like hip conformation of legs, and muscle-related traits along with meat quality [15]."
2) Comments - Line 248: extra ‘f’ in VCFtools, and 3) Comment - Line 412: missing ‘G’ on LZG.
Author response: Thank You for pointing this out. We have corrected both typographic mistakes.
Line 269: "...using VCFtools v.0.1.17 [34] with a 50 kb sliding...".
Line 428: "...whereas for the LZG breed...".
Line 445: "...among animals of the LZG breed...".
4) Comment - Lines 470-471: "Although it is slightly higher, are these values really different?"
Author response: As the reviewer has rightly pointed out, the statistical significance of differences between the inbreeding estimates calculated for LZG and LBG cattle breeds has not been given. To correctly describe the differences in the level of inbreeding between LZG and LBG animals we have included p-values from a Wilcoxon rank sum test in the text within the results section. Necessary changes have been also added in the methods section. The statistical test indicates that there is small yet significant difference in inbreeding levels between the two breeds.
We have introduced necessary changes in the text:
Line 272: "The distribution of nucleotide divergence, Tajima’s D statistic, FROH and FROH>50kb values were assessed using the Shapiro–Wilk normality test and further used to compare their significance between LBG and LZB breeds using the Wilcoxon rank sum test at a p-value threshold of 0.05."
Line 513-516: "FROH based on the sum of all ROH taken together was slightly, but significantly, higher in the LBG breed (0.197) than in the LZG breed (0.179; p=0.038). Inbreeding measures that were calculated considering only ROH that were longer than 50kb or FROH>50kb reached 0.12 for LBG and 0.10 for LZG breeds (p=0.02)."
5) Comment - Lines 484-491: Here and related to mastitis in general: there is a strong effect of environment and management. The numbers of animals with culturable bacteria could be related purely to management and less so with genetic traits. The description of the number of herds used for sampling is difficult to follow in the methods. Can a definitive statement be made in the methods about the number of sites and how many animals were of each breed from each site?
Also, can a clarification be included about the animals with cultured bacteria and somatic cell counts be indicated by number of sites or herds? - were all the mastitic cows from one herd or multiple herds?
Author response: We agree that a clear representation of this information is essential for the interpretation of the results. We have summarized information on a number of herds and cows from each herd included in the study in the table provided in Supplementary data (Table S1, Sheet2). In the case of LBG cows came from six herds (from one to 18 cows per herd). LZG animals are more often kept in smaller farms and cows included in the study came from eight different herds (from one to nine cows per herd). Animals with mastitis and cultured bacteria came from multiple herds for both breeds.
Reference to Supplementary table has been inserted in methods section:
Line 149-153: "...the dominant Rasa dams. All eight related cows were on the same farm along with other 11 cows selected for the study. Sample sets also included dam Zube and her daughters Zviedre and Zane (Figure S1). The sample set also included Zube and her daughters Zviedre and Zane, who belonged to a different herd (Figure S1). LBG animals included in the study came from six different herds (Table S1)."
Line 172-173: "Cows of the LZG breed were selected from eight herds (Table S1)."
Line 175-176: "Milk samples were obtained from 28 LBG cows and 21 LZG cows kept in six and seven herds respectively (Table S1)."
Line 548-550: ".... were used to investigate genetic traits linked to mastitis susceptibility (Table S1). Acquired genome-wide association results revealed...".
5) Comment - Line 499: figure 4E
Author response: Reference to the respective figure has been corrected.
Line 552: "... LZG breed variants (Figure 4E, Figure S13) ...".
6) Comment - Line 511: G to LZ
Author response: Thank You for pointing this out. We have corrected the typographic mistake.
Line 563: "... For the LZG breed, mastitis-associated...".
7) Comment - Line 526: alternate or reference?
Author response: We thank the reviewer for the raised question. Indeed, there was a mistake. We have corrected respective lines with the correct information from:
Line 525-527: “… were homozygous to the reference allele (T/T), whereas 9 healthy and 10 undiagnosed 525 cows were homozygous to the reference allele (TG/TG). Only one healthy and three undiagnosed cows were heterozygous (TG/T).”
to:
Line 577-579: “… 11 diseased and 6 undiagnosed cows were homozygous to the reference allele (CG/CG), whereas 9 healthy and 10 undiagnosed cows were homozygous to the alternate allele (C/C). Only one healthy and three undiagnosed cows were heterozygous (CG/C).”
8) Comment: Can a forward looking statement about the potential to preserve or conserve these breeds be added to the Conclusions? Authors indicate a need to do so, but what is the potential for success given the inbreeding, limited stock, and crossbreeding likely to be happening. Is it realistic to believe these breeds will be preserved and what will be required?
Author response: We thank the reviewer for the valuable comments. Preservation of both local breeds is indeed hindered by all these factors. Although LBG has suffered less from crossbreeding compared to LZG cattle, both breeds are critically endangered. It is likely that some proportion of the original genetic composition of both breeds has been already irreversibly lost, which raises the question of the practicality of conservational efforts. However, we believe that both breeds still have genetic value and represent part of cultural heritage. Therefore, all efforts should be made if not towards restoration of the breeds in their original state then at least to preserve the characteristic phenotype of local cattle and what remains of breed-specific genetic signature. To preserve both breeds it is necessary to continue the genetic characterization of the remaining cattles. Integration of genetic information into breeding programs would facilitate the selection of the best animals for the breeding programs. Currently, the number of breeds with genome-wide data is increasing, which provides an opportunity to select the most suitable breeds for crossbreeding, which might be necessary to retain a healthy genetic diversity, given the small number and inbreeding of local cattle. Although there are breeding programs for LZG and LBG cattle, the establishment of a "living gene bank", particularly in the case of LZG that has been scattered among several farms, would significantly improve the odds of preserving the breed. Apart from that, conservation efforts also depend on support from the state and the motivation of farmers to breed local cattle. For conservation efforts to be successful, it would be essential to continue to develop collaboration between farmers, researchers, and the state.
We have added comments on the conservation efforts of LBG and LZG breeds in the conclusion section:
Line 841-848: "However, moderate levels of heterozygosity and elevated inbreeding indices indicate an urgent need to preserve both breeds' genetic integrity. Some proportion of original genetic traits may have been already irreversibly lost, which hinders the restoration of the original genetic profile of both breeds. However, local breeds still are of genetic value and represent part of cultural heritage. To preserve these resources it would be necessary to continue the genetic characterization of the remaining cattle. Accumulation of genetic data would facilitate the selection of the most representative animals for breeding and help to choose breeds most suitable for crossbreeding which might be necessary to retain healthy genetic diversity. Highlighting genetic variants that potentially increase disease susceptibility can improve pedigree-based breeding programs by reducing the possibility of the accumulation of multiple risk alleles that compromise the fitness of LBG and LZG breeds. Genomic data could also be used to establish "living gene banks" improving the odds to preserve local cattle breeds. Apart from assisting in the management of national breeds...."

Reviewer 2 Report
General remarks
I think that most of the data written in the Results section should be presented with tables in order to increase the readability of the text and to better compare the results of different analyses.
Why you chose mastitis/SCC for your analyses?
Do you consider LBG and LZG (LB and LZ) to be true local breeds or two cow populations of local origin? In my opinion, your results show a complex origin of both LBG and LZG.
Specific comment
Lines 127-129: are the cows unrelated? If yes at which level of the pedigree (parents, grandparents, …). Are the selected animals consanguineous?
Figures S1 and S2: could you please mark the sampled animals?
Lines 275-276 FROH in Eq.2 probably should be FROH>50kb
Lines 287-293: how many animals per breed you utilized for the association analysis (GWAS?). Are the utilized samples enough for this kind of analysis?
Line 297: SCC is not a microbiological analysis.
Line 361: please change this part of the sentence (as well as between LZG, LBG, and European breeds (Figure 2B).) because there I see a partial overlap between LZG and the other breeds you used for the comparison. Most samples were well separated from the Latvian breeds and the other breeds, but the lowest part of the LZG breed samples overlapped with the other utilized samples. Have you tried to consider also PC3 considering the low relevance of both PC1 and PC2 in differentiating the samples?
Line 383: Materials and Methods did not describe the “randomly rarefying” approach. Are you sure that this reduction of the samples is representative of the whole breed?
Figure S10: some LZG samples are mixed with the LGB animals. Please add a comment.
Lines 497-498: do you performed also microbiological analyses of milk samples to identify the microbe populations?
Lines 498-500: are these (very few) markers related to the microbe population most represented in each breed? If not please comment on (and compare) these results.
Lines 693-694: “the intronic region”? you should write “an” intronic region or specify the genes where the intron is located.
Line 696: “In line with findings from recent meta-analyses,” > citation needed.
Have you checked if there are QTL for the considered traits in the genomic regions where you found the significative markers? Are there other studies in cattle about the genes you described?
Line 588: thrice > three times
Author Response
Manuscript ID: animals-2433556
Response to Reviewers
Dear Jessica Pan,
Thank you for giving us the opportunity to submit a revised version of the manuscript "Genomic characterization of two Latvian local Bos taurus breeds and identification of genetic traits associated with mastitis" for publication in the journal Animals. We appreciate You and the Reviewers for taking the necessary time and effort to review the manuscript. We are grateful for the valuable and constructive comments. These have greatly helped us to improve the quality of the current version of the manuscript. We have carefully considered the comments and incorporated necessary changes in the manuscript. Revisions are highlighted in yellow font within the manuscript. Please find below our point-by-point response to the reviewers' comments. Line numbers correspond to the revised version of the text. We hope the manuscript after careful revisions meets the expected standards to be suitable for publication in the journal Animals.
Reviewers' Comments to the Authors:
Reviewer 2
General remarks:
Remark 1: I think that most of the data written in the Results section should be presented with tables in order to increase the readability of the text and to better compare the results of different analyses.
Author response: We thank the reviewer for the suggestion. After careful consideration, we believe that there is not enough data to be tabulated, as most of the results are already presented in figures, tables, and supplementary files.
Remark 2: Why you chose mastitis/SCC for your analyses?
Author response: We thank the reviewer for the question. It is known that mastitis is the most prevalent disease in ruminant industry with major economic, hygienic and welfare implications and it persist in all animal production systems despite the implementation of improved management practices. Here in Latvia, farmers have observed that there are mastitis-resistant animals within existing herds. Therefore, we initially speculated, that due to the bottleneck effect, in some breeds the prevalence of disease associated gene alleles might be increased as similar effect has been observed in humans. Therefore, identifying potential genomic regions associated with mastitis might help to reduce the incidence of mastitis through genetic selection, which in turn is of great interest from both an economical and welfare point of view. These aspects have been described in length within Introduction section Lines: 75-103.
Remark 3: Do you consider LBG and LZG (LB and LZ) to be true local breeds or two cow populations of local origin? In my opinion, your results show a complex origin of both LBG and LZG.
Author response: Yes, we agree that both breeds have rather complex backgrounds and currently there are problems with crossbreeding, more so in the case of LZG cattle. Technically speaking both breeds can be classified as two local cattle populations, however, animals of the LZG and LBG breeds still retain characteristic phenotypes. Both breeds are officially recognized not only in Latvia, but also internationally – both have been listed in Domestic Animal Diversity Information System by the Food and Agriculture Organization. Therefore we use the term "breed" for the two phenotypically distinct populations of native cattle.
Specific comments:
1) Comment - Lines 127-129: are the cows unrelated? If yes at which level of the pedigree (parents, grandparents, …). Are the selected animals consanguineous?
Author response: In the current study we also included related animals. Both LBG and LZG cattle included half-siblings sharing the same sire or sharing the same dam as well as animals sharing grandparents. The selection of animals was limited by available resources, predominantly the total number of animals assigned to each breed, the average relatedness between the animals within the breed, and the small number of sires used for breeding. Access to animals was also somewhat influenced by the willingness of herd owners to collaborate. Due to the demographic history of the breeds, pedigree data show complex consanguineal kinship patterns, although the average relatedness of LZG cattle is particularly high. Due to the complexity of the pedigree data, we have presented this information in the form of networks in Supplementary Figures 1 and 2. We have also revised the respective information in subsection 2.1. Selection of animals and sample collection to clarify relatedness patterns among animals included in the study.
Revised text is as follows:
Line 143-153: "The sample set included seven daughters of bulls of Rudme lineage (among them three daughters of Attals Rudme), seven of Odins lineage (including five daughters of Seims Odin's and two daughters of Viksis Odins), eight of Potrimps lineage (three daughters of Vilsons Potrimps, two of Lofs Potrimps and two of Efirs Potrimps), and 14 of Ullors lineage (among them three daughters of Roko Ullors). Several LBG cows shared maternal ancestry. Eight cows descended from one of the dominant Rasa dams. All eight related cows were on the same farm along with other 11 cows selected for the study. The sample set also included Zube and her daughters Zviedre and Zane, who belonged to a different herd (Figure S1). LBG animals included in the study came from six different herds (Table S1)."
Line 159-165: "In total, samples were taken from thirteen cows of the bull Potrimps lineage (twelve daughters of dominant sire Samts), seven of the bull Gaujars lineage, three daughters of Dzilnis and eight of Darbonis (five daughters and three granddaughters) both great-grandsons of Gaujars on the maternal side. Others included the daughter of TYG bull and two daughters of the Lithuanian Light Grey bull Semis. The Gaujars lineage had two major sub-lineages represented: Aizups Lietuvietis with two granddaughters and Jumis with five granddaughters. Some of the cows also shared maternal ancestry; for example, sisters Dalija and Durbe and Dalija's female offspring Dadze (Figure S2)."
Line 172-173: "Cows of the LZG breed were selected from eight herds (Table S1)."
2) Comment - Figures S1 and S2: could you please mark the sampled animals?
Author response: To improve the clarity of the pedigree network graphs, we have highlighted sequenced animals in Figures S1 and S2 by increasing the font size of the animal names and by adding exclamation marks.
We have also adjusted respective figure legends:
Supplementary Figure 1.: "Network representing kinship among LBG cattle.
Names of sequenced animals are indicated in brackets. Dominant paternal lineages are highlighted in distinct colours and include "Ullors"... .
... All sequenced animals are indicated with exclamation marks. Pentagons represent bulls and circles represent cows. "X" indicates animals with missing pedigree information...".
Supplementary Figure 2.: "Network showing kinship of LZG animals included in sequencing analysis.
Brackets indicate animals elected for sequencing analysis. Dominant LZ paternal lineages are ...
..."X" indicates animals with missing pedigree information. Exclamation marks indicate animals elected for sequencing analysis. Pentagons represent bulls and circles represent cows...".
3) Comment - Lines 275-276: FROH in Eq.2 probably should be FROH>50kb.
Author response: We thank You for pointing this out. We have corrected the typographic mistake.
Line 302: "FROH>50kb = LROH>50kb/LAUTO (Eq.2)".
4) Comment - Lines 287-293: how many animals per breed you utilized for the association analysis (GWAS?)? Are the utilized samples enough for this kind of analysis?
Author response: The number of animals included in the association analysis was based on the resources available for the study. For association analysis, we have used a total of 53 animals, from which it was possible to collect milk samples and necessary data. The subset included 16 healthy animals and 16 animals with mastitis representing Latvian Brown cow breed, and 10 healthy and 11 animals to investigate genetic susceptibility towards mastitis in Latvian Blue cow breed. We agree that the number of animals used for this particular analysis is small and it is one of the major weaknesses of our study. Nevertheless, we believe that sharing the obtained results may be beneficial for the studies conducted by other researchers. However, as the limited number of animals could have influenced the outcome of the analysis and subsequent conclusions drawn, we have added comments regarding these limitations of the conducted association analysis in the discussion section.
The following changes were introduced in the text:
Lines 548-550: “Thus, genomic data from 16 healthy and 16 mastitic LBG cows, as well as 10 healthy and 11 mastitic LZG cows were used to investigate genetic traits linked to mastitis susceptibility (Table S1).”
Lines 823-834: "Although conducted association analysis provides insight into the genetic background of mastitis susceptibility in local cattle populations, the contribution of potential risk variants to the development of the disease should be confirmed among the rest of the animals representing the two breeds and preferably in larger cattle populations. A limited number of animals available for the analysis is the major weakness of the current analysis. Despite the measures taken to avoid erroneous signals, the small sample size along with the heterogeneous genetic background of some animals, particularly from the LZG breed due to the high admixture rate, could have increased the possibility of false positive findings as well as masked real associations. Nevertheless, considering the high complexity of genetic susceptibility to bovine mastitis, current findings may guide further search for causal genetic risk factors."
5) Comment - Line 297: SCC is not a microbiological analysis.
Author response: We thank You for pointing this out. We have corrected the title of the subsection 2.9.
Line 324: "2.9. Statistical analysis of SCC data".
6) Comment - Line 361: please change this part of the sentence (as well as between LZG, LBG, and European breeds (Figure 2B)) because there I see a partial overlap between LZG and the other breeds you used for the comparison. Most samples were well separated from the Latvian breeds and the other breeds, but the lowest part of the LZG breed samples overlapped with the other utilized samples. Have you tried to consider also PC3 considering the low relevance of both PC1 and PC2 in differentiating the samples?
Author response: As the reviewer has correctly indicated, the sentence in Line 361 "as well as between LZG, LBG, and European breeds (Figure 2B)) ..." gives inaccurate description of PCA results presented in Figure 2B. We have introduced necessary changes in the sentence to provide correct description of results presented in PCA plot.
Regarding the partial overlap between LZG and other breeds, the LZG animals have been crossbred with Holstein cattle. Animals with a higher percentage of Holstein ancestry among LZG cattle may cluster closer to other European breeds. According to the pedigree data, LBG cattle have not been crossbred with any of the European breeds represented in the study, which would explain LBG animals forming a more distinct cluster. We have also added comments on the subject in the text of the manuscript.
The text has been changed as follows:
Lines 389-394: ".... as well as between LBG and European breeds (Figure 2B). Several animals representing the LZG breed grouped close to the cluster formed by other European cattle breeds, which included Holstein. Observed clustering patterns agreed with the data on the ancestry of animals representing the two local breeds, with LZG animals sharing a higher proportion of Holstein ancestry."
Lastly, regarding the principal components (PC1, PC2, PC3) –Figure 2A shows components that best explain the variance of the LBG and LZG samples, namely PC1=13.5% and PC2=7.2%. Other PCs showed lower values, for instance PC3=6.2% and PC4=5.8%, hence they were not included in the manuscript. In a similar manner, in Figure 2B we showed those PCs that best explain the variance between LBG, LZG and other breeds obtained from NCBI SRA. The first two components were superior, e.g., PC=12.1% and PC2=7.93%, while other components were inferior, e.g., PC3=6.42%, PC4=5.8%. Given the results described above, we concluded that only PC1 and PC2 should be included in the manuscript, as they explain the largest proportion of variance between samples.
7) Comment - Line 383: Materials and Methods did not describe the “randomly rarefying” approach. Are you sure that this reduction of the samples is representative of the whole breed?
Author response: We thank the reviewer for the remark. We apologize for the missing information. We have updated Lines 278-284 with an explanation and methodology for reducing the sample set. The reason for rarefying the sample set is that we could not use the full dataset of LBG and LZG samples for interbreed comparisons because 40 samples from HLS, HRF, LMS, BBL and other breeds are not available in the NCBI SRA. Some analysis, such as Linkage disequilibrium, are sensitive to different sample sizes, therefore wherever comparisons of breed values wre performed, we employed a reduced sample set of LZG and LBG (the only exception being the admixture analysis), and these cases are clarified throughout the manuscript. We acknowledge that the use of only three samples per breed does not provide high statistical power, but we believe that this is sufficient to reflect overall genetic variation within each breed.
The newly introduced text reads as follows:
L278-284: “To compare LBG and LZG breeds to other breeds that were obtained from the NCBI SRA, the set of LBG and LZG samples was randomly reduced to three samples per breed using bcftools. An additional check was performed on randomly selected samples to ensure that they were not located close to each other in the genetic distance tree (Figure S10). Sample reduction of the LBG and LZG sample sets was done to ensure comparability with breed samples obtained from SRA (n=3 per each breed).”
8) Comment: Figure S10: some LZG samples are mixed with the LGB animals. Please add a comment.
Author response: As suggested by the reviewer, we have added comments regarding a cluster of LZG animals located among LBG animals within the genetic distance tree. Four animals of the LZG breed have originated from the same major paternal lineage and three cows were closely related (two half-sisters and sisters daughter). One of the explanations could be that there is some proportion of shared ancestry between the two local cattle breeds. The LBG animals that form clusters adjacent to the LZG cows represent two major LBG paternal lineages. The use of sires from lineage Potrimps for crossbreeding has been well documented. There is no data on an admixture of animals from LBG lineage Odins in the LZG breed. As we have mentioned in the first draft of the manuscript, it cannot be excluded that available pedigree records contain incomplete or erroneous data. Furthermore, the ancestry of local cattle has been documented in detail only with the beginning of artificial selection. It is also possible that the relatively high crossbreeding observed among LZG cattle could have affected clustering patterns based on genetic distances. We have revised the text in the results section and added suggested comments.
The revised text reads as follows:
Lines 497-502 and 505-508: "...from the pedigree information available at AIC. Four related animals of LZG breed grouped with LBG animals. All four were descended from the same paternal lineage and three were closely related (two half-sisters and sisters daughter). LBG animals forming clusters flanking group of LZG cows represent two major LBG paternal lineages. Use of sires from lineage Potrimps for crossbreeding has been well documented, but there is no data on admixture of animals from LBG lineage Odins in LZG breed. Since overall genetic distances were low, the distribution of animals within the genetic tree could have been affected by an admixture of different breeds at different proportions within ancestral lineages. Nevertheless, ancestry of local cattle has been documented in detail only with beginnings of artificial selection. One of the explanations could be that there is some proportion of shared ancestry between the two local cattle breeds. Pedigree data may be incomplete and errors in pedigree records cannot be excluded."
9) Comment - Lines 497-498: do you performed also microbiological analyses of milk samples to identify the microbe populations?
Author response: Presence of bacteria in milk samples was detected by bacterial growth in cultures. Species identification was based on morphological and biochemical characterisation of isolates, which is a routine approach for testing the quality of milk. Thorough characterisation of bacterial communities in milk samples, for example, employing 16S rRNA variable region sequencing was not performed and was outside the scope of the current study.
10) Comment - Lines 498-500: are these (very few) markers related to the microbe population most represented in each breed? If not please comment on (and compare) these results.
Author response: The presence of bacteria in milk samples was analyzed using a standard approach focused on the major mastitis-related pathogens. Both breeds had the same set of pathogens tested. The prevalence of bacteria as well as a spectrum of species differed between the LZG and LBG cows. Ten bacteria species were found in LBG milk samples, and seven were detected in milk samples from the LZG breed. Variation in the prevalence of bacteria in milk samples could be explained by exposure to different environmental factors since LZG and LBG cows were kept in different herds. Possibly there are differences in resistance towards the pathogens between the two breeds.
The text has been modified to clarify these aspects and respective comments have been added:
Lines 531-537: ".... major pathogens detected in milk. From the tested set of bacterial species six (Enterococcus faecalis, Lactococcus lactis, and Streptococcus species S.uberis, S.xylosus, S.dysgalactiae and S.simulans) were found only in LBG milk samples while three (Aerococcus viridans, Streptococcus mitis and Staphylococcus simulans) were specific to LZG samples. Variations in the prevalence of bacteria in milk samples could be due to environmental factors and differences in resistance toward pathogens between the two breeds. The prevalence of bacterial agents in...."
11) Comment - Lines 693-694: “the intronic region”? you should write “an” intronic region or specify the genes where the intron is located.
Author response: According to the reviewer's comment, location of the genetic variants has been specified by adding gene names in the respective sentence.
Changes introduced in the text were as follows:
Lines.745-747: "We found four genetic variants associated with mastitis among LBG and LZG cattle. Three were located within intergenic regions, whereas the fourth was found within the intronic region of CTSW and CADPS genes.”
12) Comment - Line 696: “In line with findings from recent meta-analyses,” > citation needed.
Author response: We have inserted the missed citation.
Lines 752: "In line with findings from recent meta-analysis, .... genes with plausible roles in the development of mastitis [15]."
13) Comment: Have you checked if there are QTL for the considered traits in the genomic regions where you found the significative markers? Are there other studies in cattle about the genes you described?
Author response: Thank You for this suggestion. Currently, we have not conducted QTL analysis and therefore we cannot comment on whether any QTL signals are mapping to the genomic regions harbouring genetic variants that might correlate with mastitis in our data. However, to address this question, we have re-checked published QTL studies for mastitis-related traits, predominantly SCC, that were found near the association signals detected in our study. We have found additional information on two of the discovered loci harbouring genes CTSW, FIBP and EFEMP2, and FEZF2, respectively. The rest of the potential candidate loci, to our knowledge, have not been described so far in the context of mastitis. We apologize for having supplied incomplete information. We have added the novel information in the respective sections of the discussion along with the matching references.
The text has been reorganized and new information added as follows:
Lines 753-766 and 771-779: "Genetic variants located within the genomic region on chromosome 29 overlapping CTSW and the two other genes found within the neighbourhood, FIBP and EFEMP2, have been reported among quantitative trait loci associated with the somatic cell count in Jersey cattle [15; 63]. The causal link between the genes representing the locus and mastitis-related traits has not yet been established. The product of CTSW, Cathepsin W, is expressed only in cytotoxic T-cells and natural killer cells, where it is involved in regulating target cell killing [64]. The direct role in cell-mediated cytotoxicity makes CTSW a likely candidate involved in the pathophysiology of mastitis. Other genes found within the neighbourhood include FIBP, EFEMP2, and CCDC85B. The second-best candidate probably is FIBP and it is also the nearest to the target SNVs. The FIBP binds fibroblast growth factor 1 (FGF1), which is involved in defence mechanisms against bacterial invasion in mastitis. The FGF1 was among the genes induced by E.coli infection of primary mammary epithelial cells in the udder [65]. Expression of FGF1 induces calcium mobilization and activation of NF-κB and other signalling pathways within immune cells with the net result of reduced inflammation [65,66]. The role that EFEMP2 could play in the development of mastitis is less clear. The results of previous GWAS studies suggested that the EFEMP2 locus is affecting cattle meat quality [68]. Another upstream-located gene that encoded transcriptional repressor CCDC85B may promote the progress of inflammation of udder tissue through mechanisms related to those of C2CD4A. Activation of CCDC85B enhances β-catenin degradation, retaining NF-κB activity and thus inhibiting the Wnt/β-catenin cascade. Inhibition of the Wnt/β-catenin pathway prevents re-organization of the cytoskeleton of epithelial cells. Preserving cytoskeleton structure maintains the integrity of the endothelial barrier and impedes bacterial invasion within the tissue [68,69]."
Lines 797-800: "... CADPS may be involved in the development of mastitis through multiple mechanisms. Yet, FEZF2 found upstream from the rs383806754 has been linked with mastitis by other studies [16; 73; 74]. Copy number variant located within the gene increased susceptibility to mastitis almost twice among cattle in the study conducted by Japanese researchers. Furthermore, FEZF2 expression was increased in the mastitic quarters of affected cows [16; 74]. The FEZF2 has been suggested to be involved in immune tolerance and defects of the gene causing ... ."
Three references have been added to the list:
Oliveira, H. R., Lourenco, D. A. L., Masuda, Y., Misztal, I., Tsuruta, S., Jamrozik, J., Brito, L. F., Silva, F. F., Cant, J. P., & Schenkel, F. S. (2019). Single-step genome-wide association for longitudinal traits of Canadian Ayrshire, Holstein, and Jersey dairy cattle. Journal of dairy science, 102(11), 9995–10011. https://doi.org/10.3168/jds.2019-16821
Ogorevc, J., Kunej, T., Razpet, A., & Dovc, P. (2009). Database of cattle candidate genes and genetic markers for milk production and mastitis. Animal genetics, 40(6), 832–851. https://doi.org/10.1111/j.1365-2052.2009.01921.x
Sugimoto, M., Fujikawa, A., Womack, J. E., & Sugimoto, Y. (2006). Evidence that bovine forebrain embryonic zinc finger-like gene influences immune response associated with mastitis resistance. Proceedings of the National Academy of Sciences of the United States of America, 103(17), 6454–6459. https://doi.org/10.1073/pnas.0601015103
14) Comments on the Quality of English Language: Line 588: thrice > three times.
Author response: Thank You for pointing this out. We have corrected the grammatical mistake.
Line 641: "... to Holstein supporting shared history, whereas three times as high a value of FST 0.175 was estimated for Yakutian cattle, an ...".

Round 2
Reviewer 2 Report
See my comments written in red characters in the attached file

Author Response
Manuscript ID: animals-2433556
Dear Academic Editor,
We are pleased to resubmit our revised manuscript entitled “Genomic characterization of two Latvian local Bos taurus breeds and identification of genetic traits associated with mastitis” for consideration of publication in the journal Animals. We confirm that this work is original, has not been previously published, and is not currently under review by any other journal.
We would like to express our sincere gratitude to You and the Reviewers for dedicating your time and expertise to reviewing our manuscript. The valuable and constructive comments provided have significantly enhanced the quality of the current version.
We have carefully considered all the comments and suggestions from the Reviewers and have made the necessary revisions in the manuscript. Attached, you will find a detailed point-by-point response addressing each of the Reviewer's comments.
We believe that the revisions have substantially improved the manuscript, and we hope that it now meets the standards for publication in MDPI Animals.
Thank You once again for your consideration, and we look forward to Your response.
Sincerely,
PhD Anda Valdovska
On behalf of all authors of this interdisciplinary work
